# Maternal H3K36 and H3K27 HMTs protect germline development via regulation of the transcription factor LIN-15B

**Chad Steven Cockrum, Susan Strome***

Department of Molecular, Cell, and Developmental Biology, University of California, Santa Cruz, Santa Cruz, United States

**Abstract** Maternally synthesized products play critical roles in the development of offspring. A premier example is the *Caenorhabditis elegans* H3K36 methyltransferase MES-4, which is essential for germline survival and development in offspring. How maternal MES-4 protects the germline is not well understood, but its role in H3K36 methylation hinted that it may regulate gene expression in primordial germ cells (PGCs). We tested this hypothesis by profiling transcripts from nascent germlines (PGCs and their descendants) dissected from wild-type and *mes-4* mutant (lacking maternal and zygotic MES-4) larvae. *mes-4* nascent germlines displayed downregulation of some germline genes, upregulation of some somatic genes, and dramatic upregulation of hundreds of genes on the X chromosome. We demonstrated that upregulation of one or more genes on the X is the cause of germline death by generating and analyzing *mes-4* mutants that inherited different endowments of X chromosome(s). Intriguingly, removal of the THAP transcription factor LIN-15B from *mes-4* mutants reduced X misexpression and prevented germline death. *lin-15B* is X-linked and misexpressed in *mes-4* PGCs, identifying it as a critical target for MES-4 repression. The above findings extend to the H3K27 methyltransferase MES-2/3/6, the *C. elegans* version of polycomb repressive complex 2. We propose that maternal MES-4 and PRC2 cooperate to protect germline survival by preventing synthesis of germline-toxic products encoded by genes on the X chromosome, including the key transcription factor LIN-15B.

*For correspondence:
sstrome@ucsc.edu

**Competing interest:** The authors declare that no competing interests exist.

## Editor's evaluation

This study provides a compelling and significant advance on the understanding of how gene regulation by the histone methyltransferase MES-4 underlies germ cell survival in *C. elegans*, with the major claims being nicely substantiated. The critical and surprising finding is that the degeneration of *mes-4* mutant primordial germ cells is due to inappropriate upregulation of genes on the silenced X chromosome, and not a failure to activate germline-expressed genes, though reduced levels of germline gene expression were observed. An X-linked target of *mes-4*, *lin-15b*, is necessary for the degeneration phenotype. The claim is compellingly supported.

## Introduction

Many critical events during early development are orchestrated by maternally synthesized gene products. Mutations in genes that encode such products in the mother can cause 'maternal-effect' phenotypes in offspring. These phenotypes are usually severe developmental defects. Maternal-effect lethal genes, which cause maternal-effect death of offspring, encode products that guide crucial events in early embryo development, such as pattern formation and embryonic genome activation (e.g., the PAR proteins in *Caenorhabditis elegans*, BICOID in *Drosophila*, and Mater in mouse) (***Nüsslein-Volhard***

*et al., 1987*; *Tong et al., 2000*; *Kemphues et al., 1988*). Maternal-effect sterile genes encode products needed for fertility of the offspring. A few genes in this category encode proteins in germ granules (e.g., PGL-1 in *C. elegans* and VASA in *Drosophila*) (*Nüsslein-Volhard et al., 1987*; *Rongo and Lehmann, 1996*; *Kawasaki et al., 1998*). Another fascinating set of genes in this category encode chromatin regulators, which are the focus of this article.

The *C. elegans* MES proteins were identified in genetic screens for maternal-effect sterile mutants, hence their name (MES for maternal-effect sterile) (*Capowski et al., 1991*). MES-2, MES-3, and MES-6 assemble into a trimeric complex that is the *C. elegans* version of Polycomb Repressive Complex 2 (PRC2) (*Xu et al., 2001*; *Bender et al., 2004*). PRC2 is a histone methyltransferase (HMT) that methylates Lys 27 on histone H3 (H3K27me) to repress genes that are packaged by those methylated nucleosomes (*Ketel et al., 2005*; *Margueron and Reinberg, 2011*; *Pengelly et al., 2013*). MES-4 is an HMT that methylates Lys 36 on H3 (H3K36me), which marks actively transcribed genes and has context-dependent roles in transcriptional regulation (*Bender et al., 2006*; *Furuhashi et al., 2010*; *Rechtsteiner et al., 2010*; *Kreher et al., 2018*). Although PRC2 and MES-4 catalyze opposing flavors of histone marking, the loss of either causes nearly identical mutant phenotypes (*Capowski et al., 1991*). Worms that inherit a maternal load of gene product but cannot synthesize zygotic product (referred to as *mes* M+Z- mutants) are fertile. Worms that do not inherit a maternal load or produce zygotic gene product (*mes* M-Z- mutants) are sterile due to death of nascent germ cells in early- to mid-stage larvae. In *mes* M-Z+ mutants, zygotically synthesized product does not rescue fertility, highlighting the critical importance of maternal product. PRC2's roles in transcriptional repression and development have been intensively studied and are well defined across species, including roles in *C. elegans* germline development (*Bender et al., 2004*; *Patel et al., 2012*; *Gaydos et al., 2014*; *Kaneshiro et al., 2019*; *Delaney et al., 2019*). In contrast, how MES-4 ensures the survival of nascent germ cells is unknown and particularly puzzling.

One possibility for MES-4 function is that maternal MES-4 promotes expression in offspring of genes required for germline development. Support for this comes from analyses of mutants that ectopically express germline genes in their soma (e.g., *mep-1, lin-15B, lin-35,* and *spr-5; met-2* mutants), and as a result have developmental defects (*Unhavaithaya et al., 2002*; *Wang et al., 2005*; *Cui et al., 2006*; *Petrella et al., 2011*; *Wu et al., 2012*; *Carpenter et al., 2021*). Concomitant loss of MES-4 from these mutants prevents ectopic expression of germline genes and restores worm health (*Wang et al., 2005*; *Cui et al., 2006*; *Petrella et al., 2011*; *Wu et al., 2012*; *Carpenter et al., 2021*). In wild-type (wt) early embryos, MES-4 and methylated H3K36 associate with genes that were transcribed in the maternal germline, regardless of whether they are transcribed in embryos (*Furuhashi et al., 2010*; *Rechtsteiner et al., 2010*). Focusing on H3K36me3, genetic tests showed that MES-4 strictly maintains pre-existing patterns of H3K36me3 and is unable to catalyze de novo H3K36me3 marking of genes (*Furuhashi et al., 2010*); the other H3K36 HMT in *C. elegans*, MET-1, like H3K36 HMTs in other systems, catalyzes de novo H3K36me3 on genes in response to transcriptional turn-on (*Kizer et al., 2005*; *Furuhashi et al., 2010*; *Kreher et al., 2018*). Taken together, these findings suggested the appealing model that in embryos maternal MES-4 maintains H3K36me3 marking of germline-expressed genes and in that way transmits an epigenetic 'memory of germline,' a developmental blueprint, to the primordial germ cells (PGCs) of offspring.

Two findings challenge the model that MES-4 somehow promotes expression of germline genes. First, among *mes-4* M-Z- mutants, hermaphrodites (with two X chromosomes) are always sterile, while males (with one X chromosome) can be fertile (*Garvin et al., 1998*). This suggested that the dosage of X-linked genes matters for the Mes-4 mutant phenotype. Second, profiling transcripts in the germlines of fertile *mes-4* M+Z- mutant hermaphrodites revealed that the most dramatic effect of losing zygotic MES-4 is upregulation of genes on the X (*Bender et al., 2006*; *Gaydos et al., 2012*). Notably, the X chromosomes are normally kept globally repressed during all stages of germline development in males and during most stages of germline development in hermaphrodites (*Kelly et al., 2002*; *Reinke et al., 2004*; *Wang et al., 2009*; *Arico et al., 2011*; *Strome et al., 2014*; *Tzur et al., 2018*), and likely as a consequence, most germline-expressed genes are located on the autosomes. These findings focused attention on the X chromosome and raised the question – what role does maternal MES-4 serve to ensure that PGCs survive and develop into a full and healthy germline?

To investigate the role of MES-4 in nascent germlines, which critically rely on maternal MES-4 for survival, and to formally test the model that MES-4 activates a germline transcription program, we

performed transcript profiling in dissected single pairs of PGCs and in dissected single sets of their descendants (which we term 'early germ cells' [EGCs]) from *mes-4* M-Z- mutant versus wt larvae. We asked whether the absence of maternal MES-4 causes nascent germlines to (1) fail to express germline genes, (2) inappropriately express somatic genes, and/or (3) inappropriately express X genes. We found evidence of all three of those effects in *mes-4* nascent germlines, the most striking of which was upregulation of many X genes. Our genetic analysis of *mes-4* mutants with different X-chromosome endowments from the oocyte and sperm demonstrated that upregulation of one or more X genes is the cause of germline death in *mes-4* M-Z- mutant larvae. This finding clarifies that MES-4's role in promoting expression of some germline genes is not its essential role in germline development and focused our analysis on the X. We identified the transcription factor LIN-15B, an X-linked gene upregulated in *mes-4* mutants, as a major cause of X misexpression and germline death in *mes-4* mutants. Our analysis also revealed that the X genes misexpressed in *mes-4* nascent germlines are oogenesis genes, suggesting that MES-4 ensures survival and proliferation of nascent germlines by repressing a toxic oogenesis program. Performing similar tests of PRC2 (*mes-3*) M-Z- mutant larvae revealed that their nascent germlines upregulate many X-linked genes in common with *mes-4* nascent germlines, and that removal of LIN-15B restores the health of their germline, as it does for *mes-4* mutants. This study revealed that maternal MES-4 and PRC2 cooperate to ensure germline survival and health in offspring by antagonizing a key X-encoded transcription factor.

## Results

### MES-4 promotes the expression of some germline genes in EGCs

MES-4 propagates an epigenetic 'memory' of a germline gene expression program during embryogenesis by maintaining H3K36me3 on genes that were previously transcribed in parental germlines (*Rechtsteiner et al., 2010*; *Furuhashi et al., 2010*; *Kreher et al., 2018*). A popular model predicts that delivery of this memory to offspring PGCs instructs them to launch a gene expression program that promotes germline proliferation and development. To test this model, we performed RNA-sequencing to determine whether PGCs from *mes-4* M-Z- (*M*aternal MES-4 minus and *Z*ygotic MES-4 minus) mutant larvae, which completely lack MES-4, fail to express a germline program (*Figure 1A*). We developed a hand-dissection strategy to isolate single sets of two PGCs from L1 larvae within 30 min for RNA-seq library preparation. We identified PGCs using a specifically and highly expressed germline marker (GLH-1::GFP). To test for gene expression defects in *mes-4* mutant PGCs as they begin to launch their germline program, we profiled transcripts in PGCs from newly hatched wt or *mes-4* M-Z- mutant L1 larvae that were fed for 30 min. We performed differential expression analysis to identify genes that are significantly downregulated (DOWN) or upregulated (UP) in *mes-4* mutant PGCs compared to wt PGCs. Our analysis identified 176 DOWN genes and 450 UP genes (*Figure 1B–D*).

To determine whether the DOWN genes include germline genes, we analyzed transcript levels and fold changes (*mes-4* vs. wt) for genes that are members of three 'germline' gene sets: (1) a 'germline-specific' set containing 168 genes that are expressed in germline tissue but not in somatic tissues, (2) a 'germline-enriched' set containing 2176 genes that are expressed at higher levels in adults with a germline compared to adults that lack a germline, and (3) a 'MES-4-bound' set containing 4132 genes that are bound by MES-4 in embryos (see 'Materials and methods' for gene sets). We found that some germline-specific genes (6 of 168 genes, or 4%), germline-enriched genes (65 of 2176 genes, or 3%), and MES-4-bound genes (124 of 4132, or 3%) are significantly DOWN in *mes-4* PGCs (*Figure 1B and E*). However, the numbers of DOWN genes for each gene set are not more than expected by chance (*Figure 1E*).

Transcription of some germline genes turns on in PGCs during embryogenesis (e.g., *Kawasaki et al., 1998*; *Subramaniam and Seydoux, 1999*; *Spencer et al., 2011*), but the major wave of zygotic genome activation in germ cells commences after L1s hatch and feed for ~1.5 hr and before the PGCs start proliferating (*Schaner et al., 2003*; *Butuči et al., 2015*). Since some gene expression defects in *mes-4* larvae may not manifest until after that time, we used our hand-dissection strategy to isolate sets of two EGCs from *mes-4* mutant and wt L2 larvae 20 hr after hatching and feeding and profiled their transcripts (*Figure 1A*, *Figure 2*). We found that *mes-4* EGCs downregulated more germline-specific genes (21 of 168, or 13%), more germline-enriched genes (247 of 2176, or 11%), and more

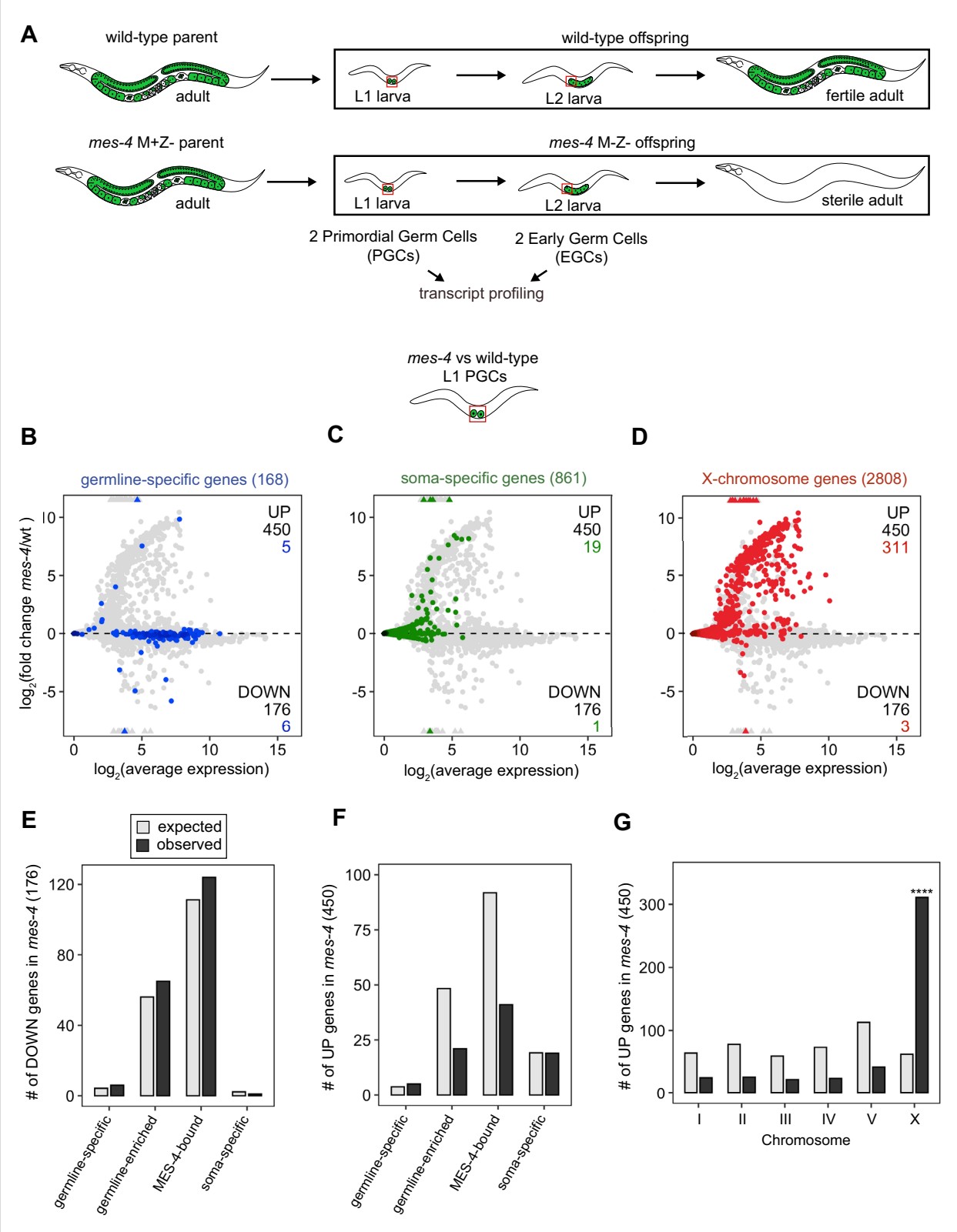

**Figure 1.** Transcriptome defects in *mes-4* M-Z- primordial germ cells (PGCs) are dominated by upregulation of genes on the X. (**A**) Cartoon illustrating the maternal-effect sterile (Mes) phenotype in *mes-4* mutants. *mes-4* M+Z- (M for maternal, Z for zygotic) mutants cannot synthesize MES-4 (Z-) but are fertile because they inherited maternal MES-4 gene product (M+). Removal of maternal MES-4 renders *mes-4* M-Z- mutant adults sterile. Germline is green, and soma is white. Transcripts were profiled from single sets of two sister PGCs and from single sets of two early germ cells (EGCs) (red boxes)

*Figure 1 continued on next page*

*Figure 1 continued*

hand-dissected from *mes-4* M-Z- (hereafter called *mes-4*) mutant and wild-type (wt) L1 and L2 larvae, respectively. (**B–D**) MA plots showing $\log_2$(average expression) versus $\log_2$(fold change) of transcript abundance for 20,258 protein-coding genes (circles) between *mes-4* and wt PGCs. Numbers of biological replicates (sets of two cells): 19 wt PGCs and 11 *mes-4* PGCs. Genes that exceeded one or both plot scales (triangles) were set at the maximum value of the scale. Genes belonging to a specific gene set are colored: (**B**) 168 germline-specific genes are blue, (**C**) 861 soma-specific genes are green, and (**D**) 2808 X-chromosome genes are red. Differentially expressed genes in *mes-4* vs. wt PGCs were identified using Wald tests in DESeq2 and by setting a q-value < 0.05 significance threshold. Numbers of all misregulated genes (black) and numbers of those in gene sets (colored) are indicated in the corners; top is upregulated (UP) and bottom is downregulated (DOWN) in *mes-4* vs. wt. (**E–G**) Bar plots showing the expected number (gray) and observed number (black) of misregulated genes that are members of the indicated gene sets. Hypergeometric tests were performed in R to test for gene-set enrichment. p-value designation is ****<1e-5. (**E**) Enrichment analyses for DOWN genes were restricted to 5858 protein-coding genes that we defined as 'expressed' (minimum average read count of 1) in wt PGCs. Gene-set sizes: germline-specific (140), germline-enriched (1867), MES-4-bound (3702), soma-specific (73). (**F, G**) Enrichment analyses for UP genes included all 20,258 protein-coding genes in the transcriptome. Gene-set sizes: germline-specific (168), germline-enriched (2176), MES-4-bound (4132), soma-specific (861), chrI (2888), chrII (3508), chrIII (2670), chrIV (3300), chrV (5084), and chrX (2808).

MES-4-bound genes (396 of 4132, or 10%) than *mes-4* PGCs (*Figure 2A, D and G*). The numbers of DOWN genes for each gene set are more than expected by chance (*Figure 2G*).

As an independent test of differential expression in PGCs, we selected three genes and performed single-molecule fluorescence in situ hybridization (smFISH) to measure and compare their transcript levels between *mes-4* and wt PGCs. We analyzed PGCs from L1 larvae that were fed for 5 hr prior to smFISH to ensure that the PGCs had sufficient time to fully turn on their germline program. Two of the genes we tested, *cpg-2* and *pgl-3*, are members of our germline-specific set and by RNA-seq analysis were DOWN or not DOWN, respectively, in *mes-4* PGCs (*Figure 1B*, *Figure 3—source data 1*). Corroborating our RNA-seq analysis, smFISH analysis showed that the transcript abundance of *cpg-2* was significantly lower in *mes-4* vs. wt PGCs, while the transcript abundance of *pgl-3* was not significantly different (*Figure 3A–C*, *Figure 3—source data 1*). The other gene we tested by smFISH, *chs-1*, is a member of our germline-enriched set and was consistently not misexpressed by RNA-seq or smFISH analysis (*Figure 3C*, *Figure 3—source data 1*). Together, our RNA-seq and smFISH analysis showed that *mes-4* nascent germ cells fail to normally express a small number of germline genes, but display near-normal expression of the majority of germline genes analyzed.

## MES-4 keeps some somatic genes off in EGCs

Chromatin regulators can protect tissue-appropriate transcription patterns by serving as a barrier to promiscuous transcription factor activity. Loss of MES-4 and PRC2 has been shown to allow misexpression of neuronal target genes upon ectopic expression of the transcription factor CHE-1 in the germline (*Patel et al., 2012*; *Seelk et al., 2016*). Moreover, loss of PRC2 activity in the *C. elegans* germline was recently linked to misexpression of some neuronal genes and conversion toward neuronal fate (*Kaneshiro et al., 2019*). Maternal MES-4 may promote offspring germline development by preventing germ cells from turning on a somatic gene expression program. To test this possibility, we examined whether UP genes in *mes-4* vs. wt PGCs and EGCs are members of a 'soma-specific' gene set that defines 861 genes expressed in soma but not in germline. We found that 19 and 64 UP genes in *mes-4* PGCs and *mes-4* EGCs, respectively, are soma-specific, the latter of which is a higher number than expected by chance (*Figure 1C and F*, *Figure 2B, E and H*). We conclude that *mes-4* germ cells begin to inappropriately express some somatic genes as they age.

## MES-4 represses genes on the X chromosome in PGCs and EGCs

Repression of the X chromosomes in the *C. elegans* germline is essential for germline health (reviewed in *Strome et al., 2014*). We found that 311 of the 2808 (11%) protein-coding X genes were UP in *mes-4* vs. wt PGCs. Strikingly, more than half of all UP genes are on the X chromosome (311 out of 450 genes), and this number is significantly higher than expected by chance (*Figure 1D and G*). *mes-4* EGCs misexpressed 568 X genes, including almost all of the X genes that were misexpressed in *mes-4* PGCs and an additional 273 X genes (*Figure 2C, F and I*). These data show that *mes-4* PGCs misexpress many genes on the X chromosome and that X misexpression becomes more severe in their descendant EGCs.

As an independent test of differential expression, we selected four X UP genes and performed smFISH to compare their transcript levels between *mes-4* vs. wt PGCs from larvae that were fed for

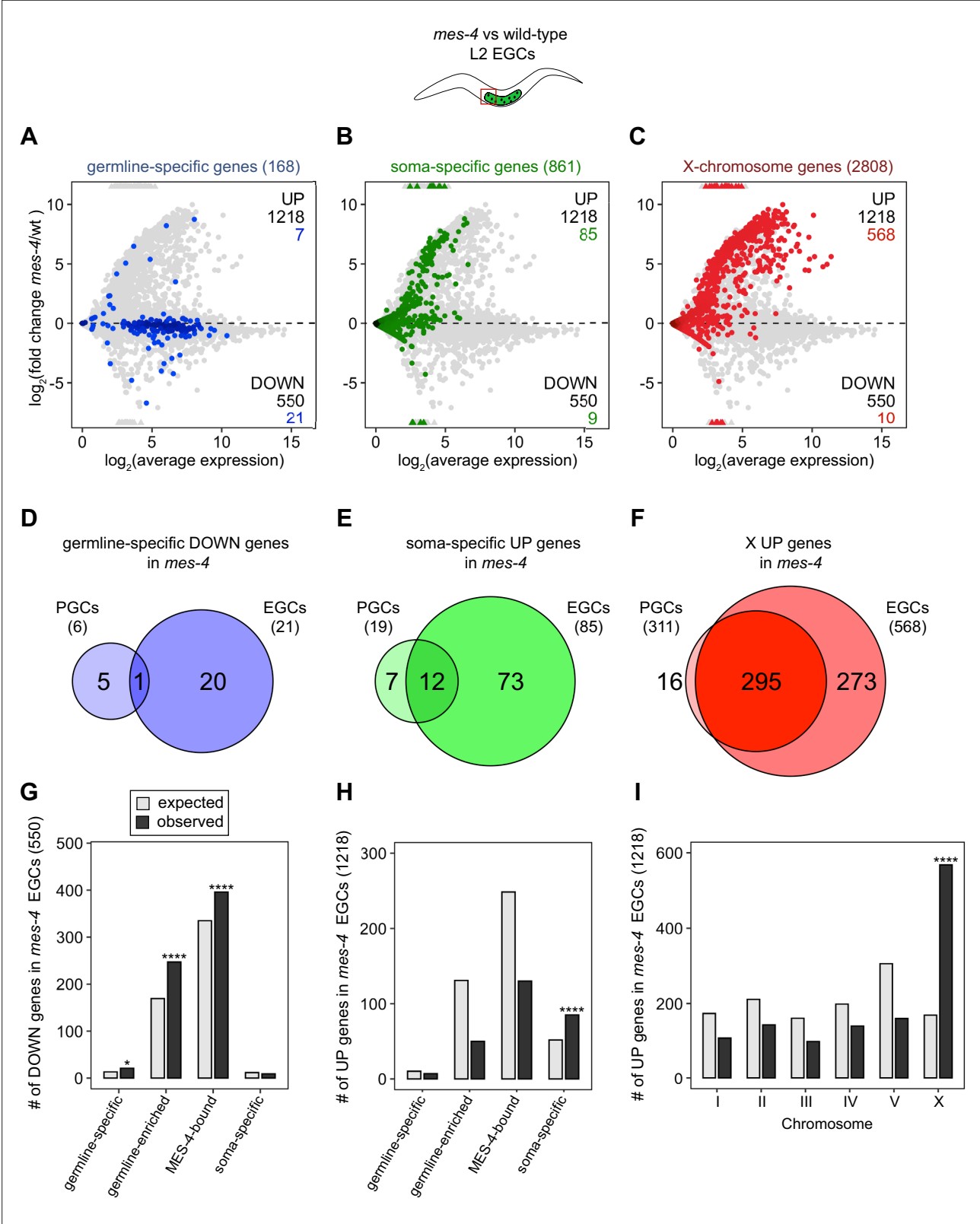

**Figure 2.** *mes-4* M-Z- early germ cells (EGCs) have more severe transcriptome defects than *mes-4* M-Z- primordial germ cells (PGCs). (**A–C**) MA plots as described for *Figure 1B–D* showing differential expression analysis for *mes-4* vs. wild-type (wt) EGCs. Numbers of biological replicates (sets of two cells): 13 wt EGCs and 15 *mes-4* EGCs. (**D–F**) Venn diagrams comparing sets of misregulated genes (*mes-4* vs. wt) between PGCs and EGCs: (**D**) germline-specific downregulated (DOWN) genes, (**E**) soma-specific upregulated (UP) genes, and (**F**) X UP genes. (**G–I**) Bar plots showing the expected number

*Figure 2 continued on next page*

*Figure 2 continued*

(gray) and observed number (black) of misregulated genes in *mes-4* EGCs that are members of the indicated gene sets. Hypergeometric tests were performed in R to test for gene-set enrichment. p-value designations are *<0.01 and ****<1e-5. (**G**) Enrichment analyses for DOWN genes were restricted to 6682 protein-coding genes that we defined as 'expressed' (minimum average read count of 1) in wt EGCs. Gene-set sizes: germline-specific (143), germline-enriched (1857), MES-4-bound (3675), soma-specific (126). (**H, I**) Enrichment analyses for UP genes included all 20,258 protein-coding genes in the transcriptome. Gene-set sizes: germline-specific (168), germline-enriched (2176), MES-4-bound (4132), soma-specific (861), chrI (2888), chrII (3508), chrIII (2670), chrIV (3300), chrV (5084), and chrX (2808).

The online version of this article includes the following figure supplement(s) for figure 2:

**Figure supplement 1.** Analysis of features of misregulated genes in *mes-4* M-Z- primordial germ cells (PGCs) and early germ cells (EGCs).

**Figure supplement 2.** Differential expression analysis of downregulated (DOWN) genes in *mes-4* primordial germ cells (PGCs) and early germ cells (EGCs) by expression decile in wild type (wt).

---

5 hr. smFISH analysis showed that all four X genes had higher transcript abundance in *mes-4* PGCs than in wt PGCs (*Figure 3D–F*), corroborating our transcriptome analysis (*Figure 3—source data 1*). These data reveal that the X chromosome is the primary focus of MES-4 regulation in PGCs.

## Misexpression of genes on the X chromosome(s) causes germline death in *mes-4* mutants

Since X misexpression is the largest transcriptome defect in *mes-4* PGCs, we hypothesized that misexpression of the two X chromosomes in germlines of *mes-4* mutant hermaphrodites causes germline death. To test our hypothesis, we asked whether *mes-4* mutant males, which inherit only a single X chromosome (typically from the oocyte), have healthy germlines. We live imaged wt and *mes-4* mutant M-Z-hermaphrodites and males that express a germline-specific reporter (GLH-1::GFP) and scored their germline health qualitatively as either 'absent/tiny' germline, 'partial' germline, or 'full' germline. All live-imaged *mes-4* mutant hermaphrodites lacked a germline (*Figure 4A*). In contrast, some *mes-4* mutant males that inherited their single X from an oocyte ($X^{oo}$ males) had either a partial or full germline (21 and 4%, respectively). Since X chromosomes turn on during oogenesis (*Kelly et al., 2002*; *Arico et al., 2011*; *Tzur et al., 2018*), $X^{oo}$ males inherited an X with a history of expression. Using a *him-8* mutant, we generated wt and *mes-4* mutant males that instead inherited their X from the sperm ($X^{sp}$ males), which has a history of repression because the X was not turned on previously during spermatogenesis. We tested whether *him-8; mes-4* mutant $X^{sp}$ males that inherited a single X with a history of repression have healthier germlines than *mes-4* mutant $X^{oo}$ males that inherited a single X with a history of expression. Strikingly, 67% of *him-8; mes-4* mutant $X^{sp}$ males made full germlines compared to only 4% of *mes-4* mutant $X^{oo}$ males.

A new and powerful genetic tool uses *gpr-1* overexpression to generate hermaphrodite worms that form a germline entirely composed of two genomes inherited from the sperm or rarely two genomes inherited from the oocyte (*Besseling and Bringmann, 2016*; *Artiles et al., 2019*). GPR-1 is a microtubule force regulator, whose overexpression in a newly fertilized embryo causes excess microtubule pulling forces on the egg and sperm pronuclei, which can prevent those pronuclei from fusing, in turn causing premature segregation of the maternal and paternal sets of chromosomes. In the two-cell embryo, the AB blastomere inherits chromosomes from one parent and the P1 (germline blastomere) inherits chromosomes from the other parent. Since the egg and sperm genomes each replicate in the one-cell embryo, all cells inherit two full genomes derived entirely from one gamete. Using this tool, we tested whether *mes-4* mutant hermaphrodites whose germline inherited two sets of sperm genomes, including 2 X chromosomes with a history of repression and 10 autosomes (called $X^{sp}X^{sp}$ hermaphrodites), have healthier germlines than *mes-4* mutant hermaphrodites whose germline inherited two sets of oocyte genomes, including 2 X chromosomes with a history of expression and 10 autosomes (called $X^{oo}X^{oo}$ hermaphrodites). While all *mes-4* mutant $X^{oo}X^{oo}$ hermaphrodites lacked a germline, some *mes-4* mutant $X^{sp}X^{sp}$ hermaphrodites had a partial or full germline (32 and 18%, respectively) (*Figure 4A*). Our combined genetic analysis demonstrates that misexpression of one or more X genes causes germline death in *mes-4* mutants. It also underscores that *mes-4* mutant PGCs can launch a transcription program that supports germline development.

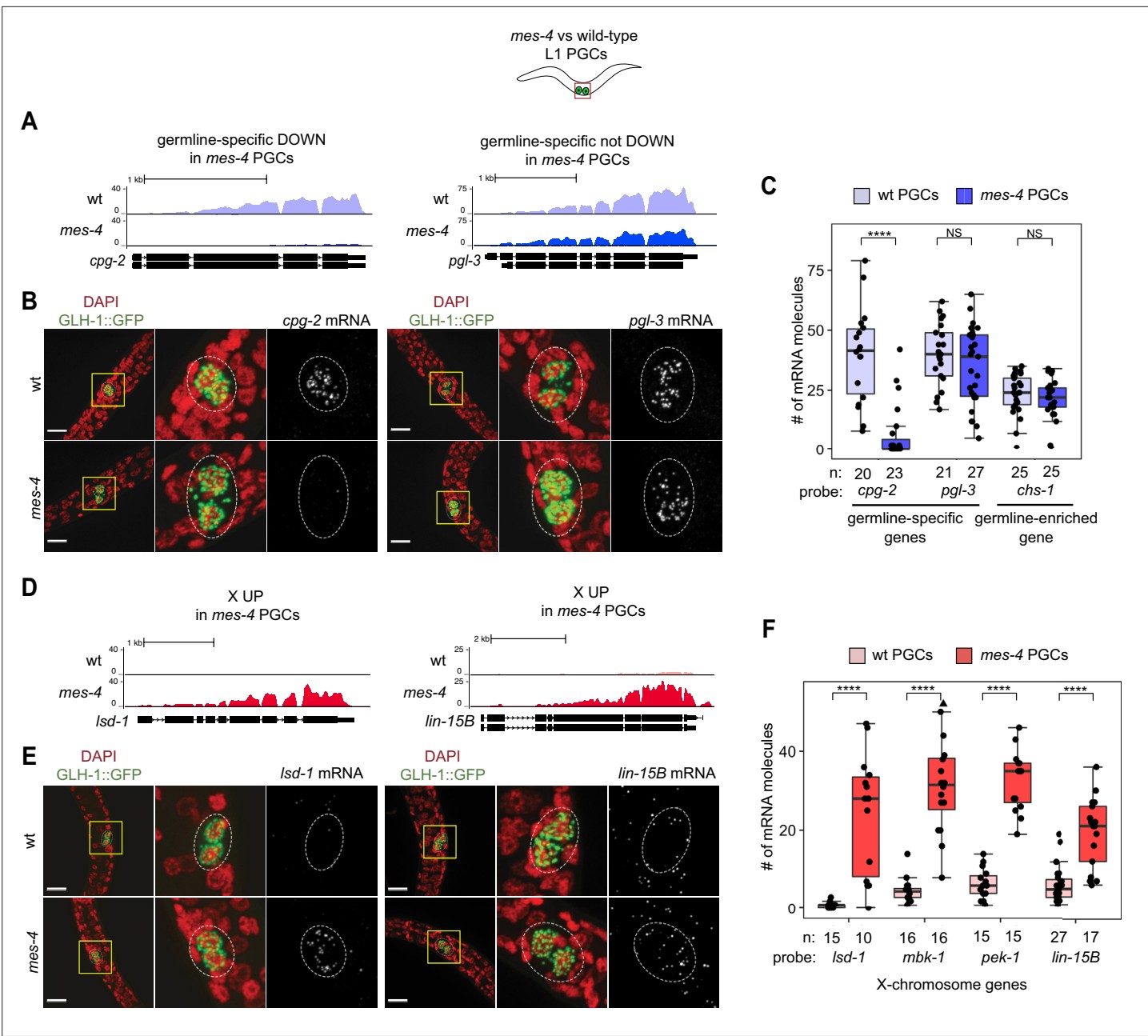

**Figure 3.** Transcript quantification in *mes-4* M-Z- mutant primordial germ cells (PGCs) by single-molecule fluorescence in situ hybridization (smFISH) corroborates RNA-seq results. (**A, D**) UCSC Genome Browser images showing average sequencing read coverage over Ensembl gene models for wild-type (wt) PGCs (top track) and *mes-4* PGCs (bottom track). (**B, E**) Representative maximum-intensity Z-projection images of smFISH experiments in L1 larvae collected after 5 hr of feeding. DAPI-stained nuclei are red. GLH-1::GFP is green. The dashed lines circumscribe PGCs marked by GLH-1::GFP. The second and third images in each set are zoomed insets of the yellow box in the first image. Foci in the mRNA channel (third image in each set) represent individual transcripts. Scale bars are 10 µm. (**A, B**) Two germline-specific genes: *cpg-2* (left) and *pgl-3* (right). *cpg-2* was downregulated (DOWN) and *pgl-3* was not DOWN in transcript profiling of *mes-4* vs. wt PGCs. (**D, E**) Two X-linked UP genes in transcript profiling of *mes-4* vs. wt PGCs and/or early germ cells (EGCs): *lsd-1* (left) and *lin-15B* (right). (**C, F**) Transcript quantification in smFISH 3D images of PGCs. Each circle represents one quantified image. The number of quantified images for each combination of probe and genotype is indicated. Boxplots show the median, the 25th and 50th percentiles (boxes), and the 2.5th and 97.5th percentiles (whiskers). Mann–Whitney tests were used to compare a gene's transcript counts between *mes-4* and wt PGCs. p-value designations are NS >0.01 and ****<1e-5. (**C**) Quantification of *cpg-2*, *pgl-3*, and *chs-1* transcripts. *chs-1* is in the germline-enriched gene set and is not DOWN in transcript profiling of *mes-4* vs. wt PGCs and EGCs. (**F**) Quantification of *lsd-1*, *mbk-1*, *pek-1*, and *lin-15B* transcripts, four X-linked UP genes in transcript profiling of *mes-4* vs. wt PGCs and/or EGCs.

The online version of this article includes the following source data for figure 3:

**Source data 1.** Comparison of single-molecule fluorescence in situ hybridization (smFISH) and transcript profiling data.

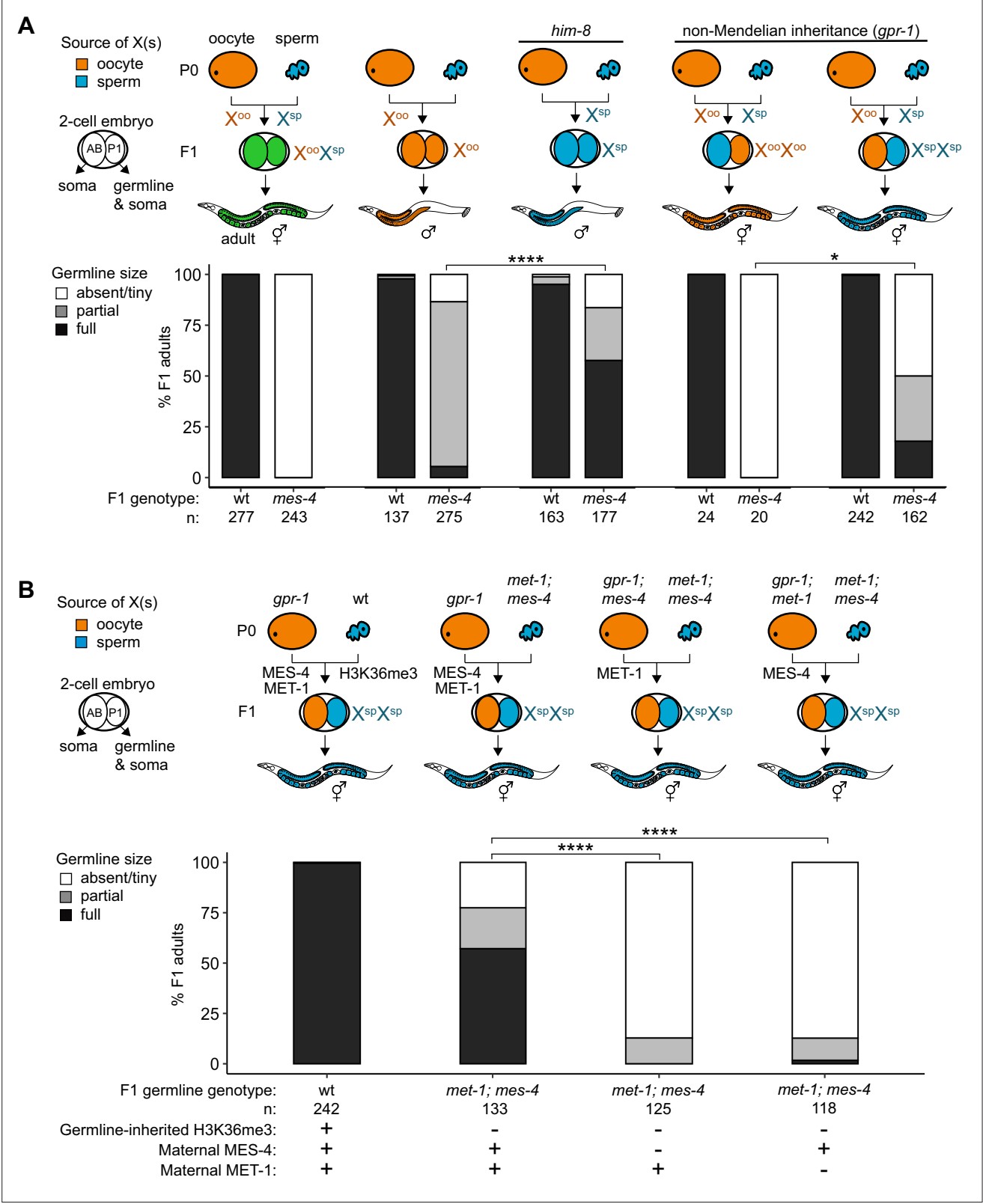

**Figure 4.** Maternally loaded MES-4 promotes germline development by repressing the X chromosomes independently from transmitting H3K36me3 across generations. (**A, B**) Bar plots showing distributions of germline size (absent/tiny, partial, and full) in worms with different X-chromosome compositions in their germline. Numbers of scored F1 offspring and the genotype of their germlines are indicated (*mes-4* indicates *mes-4* M-Z-). Two-cell embryos contain AB (left) and P1 (right) blastomeres. AB generates some somatic tissues; P1 generates the germline and some somatic tissues.

*Figure 4 continued on next page*

*Figure 4 continued*

Orange, blue, and green indicate X-chromosome compositions: orange is only oocyte-inherited X(s), blue is only sperm-inherited X(s), and green is one oocyte-inherited X and one sperm-inherited X. To generate 'non-Mendelian' F1 hermaphrodite offspring that inherited two genomes and therefore two Xs from one gamete, fathers were mated with mothers that carried a mutation in *gpr-1*. Two-tailed Fisher's exact tests were used to test whether the proportion of F1 adults with a full-sized germline significantly differed between samples. p-value designations are *<0.01 and ****<1e-5. (**A**) To generate F1 male offspring that inherited their single X from the sperm, parents were mated that carried the *him-8(e1489)* mutation, which causes X-chromosome nondisjunction during oogenesis in the hermaphrodite. (**B**) Presence or absence of sperm-inherited H3K36me3 marking, maternal MES-4, and maternal MET-1 in the germline of F1 offspring are indicated in the schematic of each cross and below each bar.

The online version of this article includes the following figure supplement(s) for figure 4:

**Figure supplement 1.** Genetic strategies to generate and identify F1 offspring that inherited different X-chromosome endowments from parents.

**Figure supplement 2.** *mes-4* M-Z+ X^sp X^sp mutants do not have healthier germlines than *mes-4* M-Z- X^sp X^sp mutants.

**Figure supplement 3.** Further fertility analyses of *mes-4* M-Z- mutants that inherited their X(s) from the sperm.

## MES-4 promotes germline health independently from its role in transmitting H3K36me3 patterns across generations

Transmission of epigenetic information across generations can impact the health of offspring. We hypothesized that maternally loaded MES-4's role in transmitting H3K36me3 patterns from parents to offspring is essential for X repression and offspring germline development. If so, then transmission of parental chromosomes lacking H3K36me3 to offspring should cause their germline to die even if they received maternal MES-4. To test our hypothesis, we used the *gpr-1* genetic tool and the GLH-1::GFP germline marker to generate F1 adult offspring whose PGCs inherited two H3K36me3(-) genomes from the sperm and either did or did not inherit maternal MES-4. To completely remove H3K36me3 from sperm chromosomes, we used *met-1; mes-4* (M+Z-) double null mutant P0 fathers, which lack the activities of both H3K36me HMTs in *C. elegans*, MET-1 and MES-4. We found that over half (57%) of F1 adult offspring had a full germline if their PGCs inherited two H3K36me3(-) genomes from the sperm and maternal MES-4 (**Figure 4B**). In contrast, 0% of F1 adult offspring had a full germline if their PGCs inherited two H3K36me3(-) genomes from the sperm and did not inherit maternal MES-4 (**Figure 4B**). This result shows that maternally loaded MES-4 is critical for offspring germline development, but that its critical role is not to transmit H3K36me3 patterns from parents to offspring.

Our finding that F1 offspring can be fertile if their germline inherited two H3K36me3(-) genomes and maternal MES-4 encouraged us to further test the role of H3K36me3 marking in germline development. One explanation for our F1 finding is that maternally loaded H3K36 HMTs can newly establish sufficient levels of H3K36me3 marking on inherited H3K36me3(-) chromosomes to repress X genes and promote germline development. If so, F1 offspring whose germline inherited two H3K36me3(-) genomes and cannot generate new H3K36me3 marking should be sterile. MES-4 cannot catalyze de novo H3K36me3 marking on H3K36me3(-) chromosomes, but the other H3K36 HMT, MET-1, can do so in response to transcriptional turn-on, like H3K36 HMTs in other species (**Furuhashi et al., 2010**; **Kreher et al., 2018**). We tested whether F1 offspring whose PGCs inherited two H3K36me3(-) genomes from sperm and maternal MES-4, but not maternal MET-1, and therefore lacked the ability to catalyze de novo H3K36me3, can make a healthy germline. We found that removal of maternal MET-1 caused almost all F1 offspring whose PGCs inherited two H3K36me3(-) genomes from sperm to lack a germline. Therefore, maternal loads of both H3K36 HMTs are required for F1 offspring whose PGCs inherited two H3K36me3(-) genomes from sperm to make a germline. These findings suggest that newly established H3K36me3 marking of H3K36me3(-) chromosomes by maternal MET-1 and maintenance of that marking by maternal MES-4 are sufficient to maintain X repression in PGCs and promote germline survival. We speculate that de novo H3K36me3 marking of autosomes maintains X repression since most X genes normally do not turn on in the embryonic germline and therefore would not get marked de novo by MET-1 (**Spencer et al., 2011**).

## LIN-15B causes X misexpression in the germline of *mes-4* M+Z- adults

MES-4 levels are low on the X chromosome(s) (**Bender et al., 2006**; **Rechtsteiner et al., 2010**; **Furuhashi et al., 2010**; **Gaydos et al., 2012**; **Kreher et al., 2018**). We therefore hypothesized that MES-4 represses X genes indirectly by regulating the expression or activity of one or more downstream factor(s). For example, MES-4's activity on autosomes may repress X genes by concentrating

a transcriptional repressor onto the X or by sequestering a transcriptional activator away from the X (*Gaydos et al., 2012*; *Cabianca et al., 2019*; *Georgescu et al., 2020*). Several lines of evidence made the THAP transcription factor LIN-15B a strong candidate for causing X misexpression in germlines that lack MES-4. First, our analysis of publicly available LIN-15B ChIP data from whole embryos and larvae found that LIN-15B targets the promoter of many X genes that are repressed by MES-4 in PGCs and/or EGCs (*Figure 5—figure supplement 1*). Second, *lin-15B* is X-linked and UP in *mes-4* vs. wt PGCs (*Figure 3D–F*, *Figure 3—source data 1*). Third, LIN-15B has been reported to promote expression of X genes in PGCs and adult germlines (*Lee et al., 2017*; *Robert et al., 2020*). Finally, *mes-4* and *lin-15B* genetically interact in somatic cells to control expression of germline genes such as genes encoding P-granule components (*Petrella et al., 2011*).

We hypothesized that LIN-15B causes misexpression of X genes in germlines that lack MES-4. To test our hypothesis, we used RNA-seq to determine whether germlines dissected from *mes-4* M+Z-; *lin-15B* M-Z- double mutant adults have reduced levels of X misexpression compared to germlines dissected from *mes-4* M+Z- single mutant adults. Our differential expression analyses showed that *mes-4* M+Z-; *lin-15B* M-Z- adult germlines upregulated considerably fewer X genes (112 X genes) than *mes-4* M+Z- adult germlines (367 X genes) (*Figure 5A*, *Figure 5—figure supplement 2*). Furthermore, the 367 X UP genes in *mes-4* M+Z- adult germlines had closer-to-wild-type transcript levels in *mes-4* M+Z-; *lin-15B* M-Z- adult germlines (*Figure 5B*), and 323 of those X UP genes were scored as X DOWN genes in *mes-4* M+Z-; *lin-15B* M-Z- compared to *mes-4* M+Z- (*Figure 5C*, *Figure 5—figure supplement 2C*). Our results show that LIN-15B is responsible for much of the X misexpression in *mes-4* M+Z- adult germlines.

## LIN-15B causes sterility in *mes-4* M-Z- mutants

Since LIN-15B causes X misexpression in *mes-4* M+Z- adult germlines, we hypothesized that removal of LIN-15B would allow *mes-4* M-Z- mutants to make healthier germlines. We compared germline health in *mes-4* M-Z- mutant F1 adult offspring that had maternal and zygotic LIN-15B (*lin-15B* M+Z+), lacked maternal LIN-15B (*lin-15B* M-Z+), lacked zygotically synthesized LIN-15B (*lin-15B* M+Z-), or lacked both (*lin-15B* M-Z-). All F1 offspring inherited two genomes from the sperm. We found that removal of either maternal LIN-15B, zygotic LIN-15B, or both increased the number of *mes-4* M-Z- $X^{sp}X^{sp}$ offspring with a full-sized germline (*Figure 5D*). Notably, loss of maternal LIN-15B caused better recovery of germline health than loss of zygotic LIN-15B, and loss of both had an additive effect. Strikingly, 88% of *mes-4* M-Z-; *lin-15B* M-Z- $X^{sp}X^{sp}$ germlines were full-sized. We conclude that both maternal and zygotic sources of the transcription factor LIN-15B contribute to germline loss in *mes-4* M-Z- mutants.

Removal of LIN-15B may only allow *mes-4* M-Z- mutant hermaphrodites to make a full-sized germline if that germline inherited two X chromosomes with a history of repression (from sperm), which by itself improves germline health in *mes-4* M-Z- mutants (*Figure 4A*). We analyzed the impact of loss of LIN-15B on germline health in $X^{oo}X^{sp}$ mutants that inherited one of their two X chromosomes with a history of expression (from the oocyte). We found that 29% of *mes-4* M-Z-; *lin-15B* M-Z- $X^{oo}X^{sp}$ adult mutant hermaphrodites made a full-sized germline compared to 0% of *mes-4* M-Z-; *lin-15B* M+Z+ $X^{oo}X^{sp}$ adult mutant hermaphrodites (*Figure 5—figure supplement 3*). This finding demonstrates that removal of LIN-15B can even allow *mes-4* M-Z- $X^{oo}X^{sp}$ mutants to make a full-sized germline.

To investigate whether other factors contribute to sterility in *mes-4* M-Z- mutants, we identified candidate genes that met one or more of three criteria: (1) they are X UP genes in *mes-4* PGCs and/or EGCs, (2) there is evidence of them binding to the promoter region of at least 25% of X UP genes, and (3) they target a DNA motif that is enriched in the promoter of X UP genes (*Figure 5—figure supplement 1*). We identified 20 top candidates based on the above criteria plus four histone acetyltransferases (HATs) that are involved in transcriptional activation and tested whether their depletion by RNAi causes *mes-4* M-Z- mutants to make a healthier germline. Of the 20 genes tested, only RNAi depletion of LIN-15B caused *mes-4* M-Z- mutants to make a healthier germline (*Figure 5—figure supplement 1*). We conclude that LIN-15B is a major contributor to germline death in *mes-4* M-Z- mutants.

## MES-4 keeps an oogenesis program off in the nascent germline

A previous study found that LIN-15B promotes expression of an oogenesis program in *nanos* mutant PGCs, which causes germline death in *nanos* mutants (*Lee et al., 2017*). We wondered whether

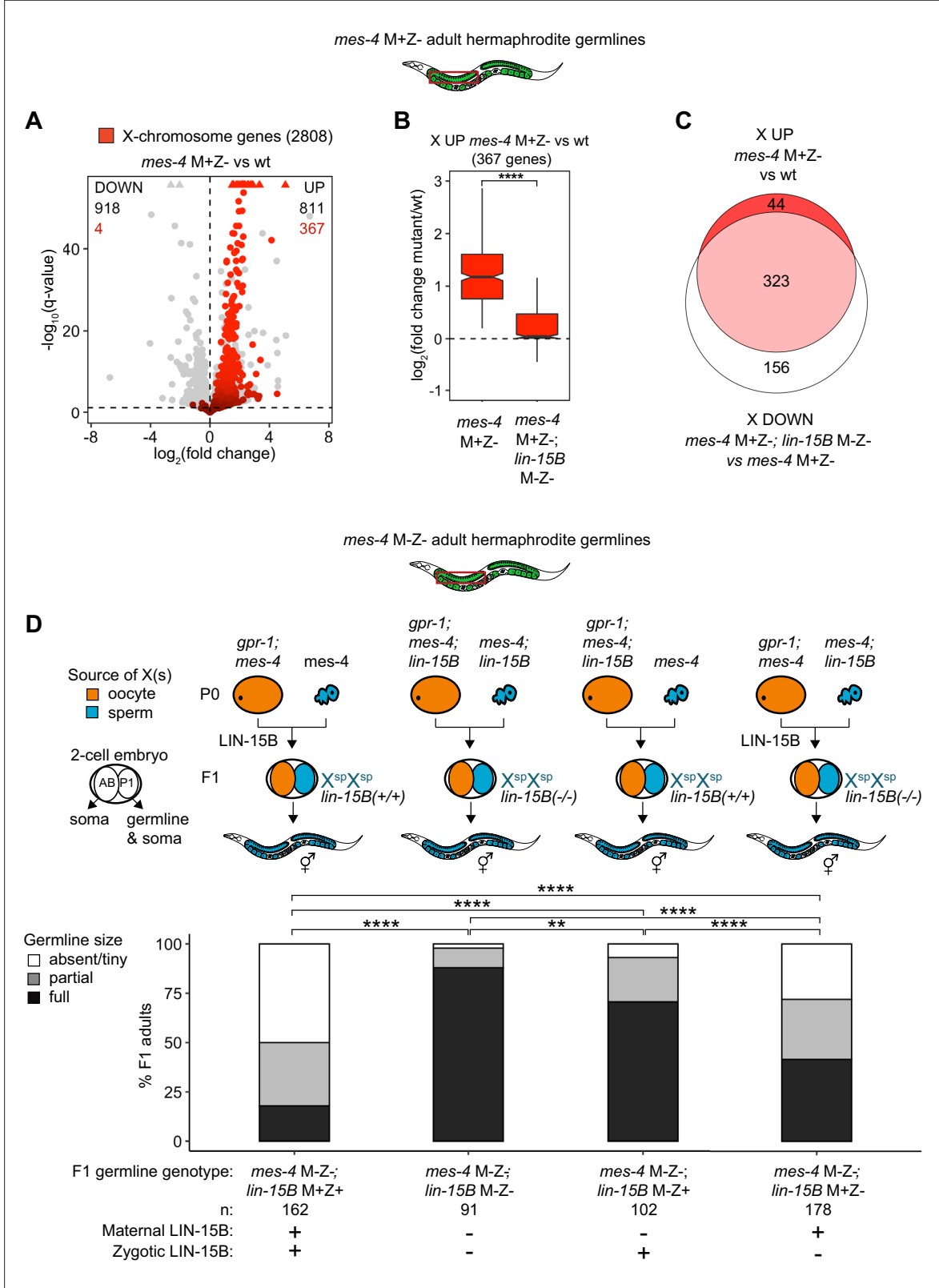

**Figure 5.** Loss of LIN-15B reduces X misexpression in *mes-4* M+Z- adult germlines and suppresses germline death in *mes-4 M-Z-* mutants. (**A**) Volcano plot showing log$_2$(fold change) of transcript abundance and significance [-log$_{10}$(q-value)] for 20,258 protein-coding genes (circles) between germlines dissected from *mes-4* M+Z vs. wild-type (wt) adults. Genes that exceeded the plot scale (triangles) were set at the maximum value of the scale. X-chromosome genes (2808) are red. Genes above the horizontal line (q-value of 0.05) are considered significantly misregulated. The number of all

*Figure 5 continued*

misregulated protein-coding genes (black) and the number of those that are X-linked (red) are indicated in the corners; left is downregulated (DOWN) and right is upregulated (UP) in *mes-4* M+Z vs. wt. (**B**) Boxplots showing log$_2$(fold change) in transcript abundance for the 367 X UP genes in *mes-4* M+Z- vs. wt germlines between *mes-4* M+Z- vs. wt germlines and between *mes-4* M+Z-; *lin-15B* M-Z- vs. wt germlines. Boxplots show the median, the 25th and 50th percentiles (boxes), and the 2.5th and 97.5th percentiles (whiskers). Waists around the median indicate 95% confidence intervals. Mann–Whitney tests were used to compare samples. (**C**) Venn diagram comparing the 367 X UP genes in *mes-4* M+Z vs. wt and the 479 X DOWN genes in *mes-4* M+Z-; *lin-15B* M-Z- vs. *mes-4* M+Z- germlines. (**D**) Bar plots as described in the legend of *Figure 4*. Genotypes of hermaphrodite and male parents are indicated at the top. All scored F1 offspring were non-Mendelian segregants (caused by the *gpr-1* mutation in mother worms) whose germline inherited two genomes and therefore two Xs from the sperm. The F1 germline's genotype with respect to *lin-15B* is indicated to the right of the two-cell embryos; '+' is wild-type allele, '-' is null allele. The presence or absence of maternal LIN-15B and zygotic LIN-15B in the germline of F1 offspring is indicated in the schematic of each cross and below each bar. Two-sided Fisher's exact tests were used to test whether the proportion of F1 adults with a full-sized germline significantly differed between samples. p-value designations are \*\*<0.001 and \*\*\*\*<1e-5.

The online version of this article includes the following figure supplement(s) for figure 5:

**Figure supplement 1.** Identification and testing of candidate transcription factors for a role in causing sterility of *mes-4* M-Z- mutants.

**Figure supplement 2.** Further analysis of how LIN-15B impacts the transcriptome of *mes-4* M+Z- dissected adult germlines.

**Figure supplement 3.** Removal of LIN-15B improves germline health in *mes-4* and *mes-3* M-Z- X°°X$^{sp}$ mutant hermaphrodites.

nascent germ cells in *mes-4* mutants also misexpress an oogenesis program, and if so, whether that misexpression is mediated by LIN-15B. We first compared our set of UP genes in *mes-4* PGCs and/or EGCs to a set of 1671 'oogenesis' genes defined as being expressed at higher levels in dissected adult oogenic germlines than in dissected adult spermatogenic germlines (*Ortiz et al., 2014*). Since UP genes are enriched for X genes, we analyzed X genes and autosomal genes separately. We found that 350 of the 584 X UP genes (60%) and 179 of the 712 autosomal UP genes (25%) are oogenesis genes, and that both proportions are higher than expected by chance (*Figure 6A–D*). We tested two other germline gene sets: 'spermatogenesis' genes, which are expressed at higher levels in dissected adult spermatogenic germlines than in dissected oogenic germlines, and 'gender-neutral' genes, which are expressed at similar levels in both germlines (*Ortiz et al., 2014*). Neither of these two gene sets displayed the same striking enrichment as oogenesis genes (*Figure 6C and D*). We conclude that *mes-4* nascent germlines misexpress an oogenesis program involving many X and autosomal genes, which may interfere with the ability of mutant PGCs to proliferate.

We hypothesized that LIN-15B binds to and causes misexpression of oogenesis genes in *mes-4* nascent germlines. This predicts that our sets of X UP genes and autosomal UP genes in *mes-4* nascent germlines should be enriched for oogenesis genes that are targeted by LIN-15B. We defined 4120 LIN-15B-targeted genes as those that have evidence of LIN-15B binding to their promoter in LIN-15B ChIP data from whole embryos and larvae. We found that our sets of X UP genes and autosomal UP genes in *mes-4* PGCs and/or EGCs are both enriched for LIN-15B-targeted oogenesis genes (*Figure 6A–D*). Strikingly, when considering just X genes, almost all LIN-15B-targeted oogenesis genes are UP in *mes-4* (165 out of 192 genes, or 86%). This percentage is much lower when considering just autosomal genes (67 out of 468 genes, or 14%). We speculate that LIN-15B causes toxic misexpression of oogenesis genes, mainly X-linked oogenesis genes, in *mes-4* nascent germlines, which ultimately causes those germlines to die.

## MES-4 cooperates with the chromatin regulator PRC2 to repress X genes

In addition to MES-4, the maternally loaded H3K27 HMT PRC2, composed of MES-2, MES-3, and MES-6, promotes germline survival and development by repressing genes on the X chromosome (*Gaydos et al., 2012*; *Gaydos et al., 2014*). To test whether MES-4 and PRC2 cooperate to protect germline health by repressing a similar set of X genes, we compared transcript profiles in wt, *mes-3* M-Z-, and *mes-4* M-Z- PGCs and EGCs. In principal component analysis (PCA), the top two principal components captured 41% of the variance across all samples and clustered *mes-4* and *mes-3* mutant samples together by germline stage and away from wt samples (*Figure 7A*). Using differential expression analysis, we identified 354 X UP genes in *mes-3* vs. wt PGCs and 443 X UP genes in *mes-3* vs. wt EGCs. We found that stage-matched *mes-3* and *mes-4* samples upregulated a highly similar set of X genes, many of which are oogenesis genes (*Figure 7B and C*). Next, we compared log$_2$(fold change) (mutant vs. wt) of misregulated X genes between *mes-4* and *mes-3* PGCs and between *mes-4* and

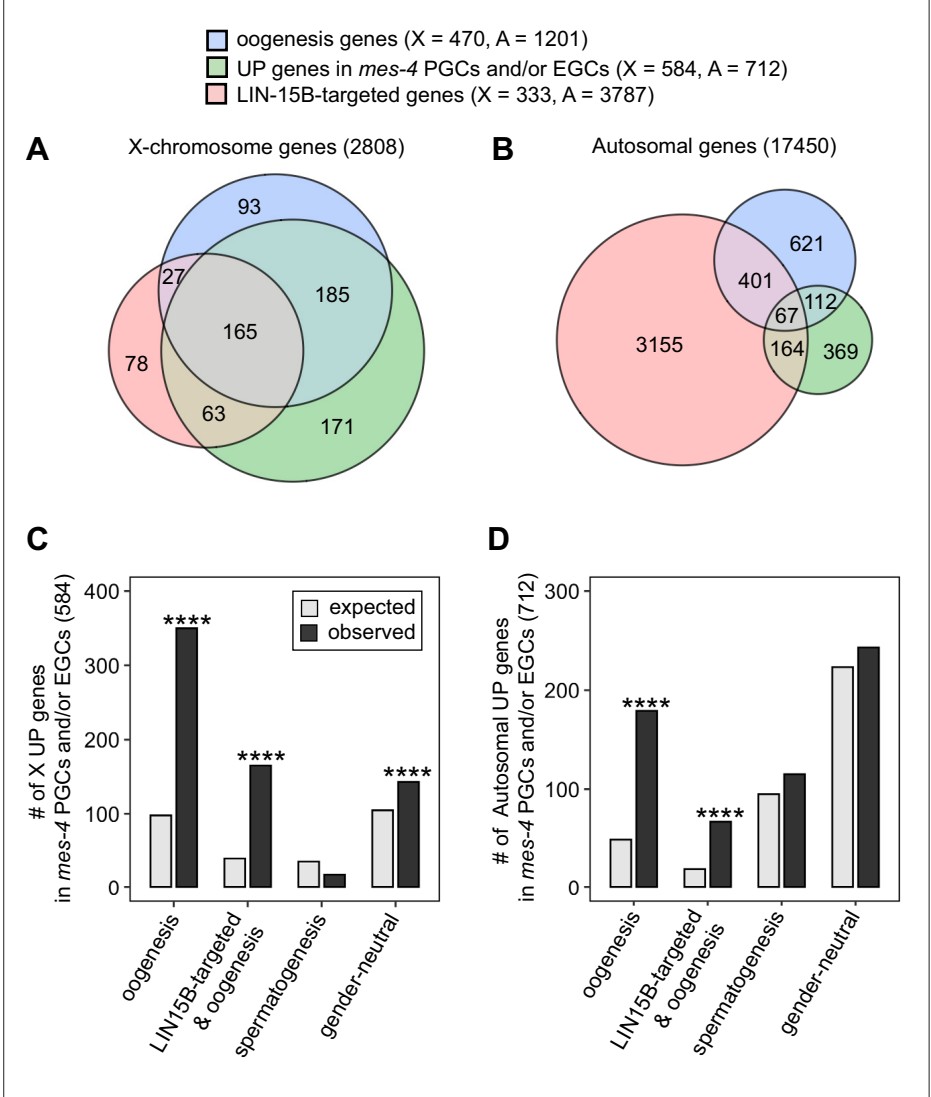

**Figure 6.** *mes-4* M-Z- nascent germlines misexpress oogenesis genes. (**A, B**) Venn diagrams comparing oogenesis genes, upregulated (UP) genes in *mes-4* primordial germ cells (PGCs) and/or early germ cells (EGCs), and LIN-15B target genes on (**A**) the X chromosome and (**B**) the autosomes. (**C, D**) Bar plots showing the expected number (gray) and observed number (black) of misregulated genes that are members of the indicated gene sets for (**C**) the X chromosome and (**D**) the autosomes. Hypergeometric tests were performed in R to test for gene-set enrichment. p-value designation is ****<1e-5. Gene-set sizes for the X chromosome: oogenesis (470), LIN-15B-targeted and oogenesis (192), spermatogenesis (171), and gender-neutral (503). Gene-set sizes for the autosomes: oogenesis (1201), LIN-15B-targeted and oogenesis (468), spermatogenesis (2327), and gender-neutral (5470).

The online version of this article includes the following figure supplement(s) for figure 6:

**Figure supplement 1.** X UP genes have enriched transcript abundance during oogenesis compared to during pre-gametic stages.

---

*mes-3* EGCs. We found a positive, albeit small, correlation between PGCs (0.22 Spearman's correlation coefficient) and a stronger correlation between EGCs (0.44 Spearman's correlation coefficient) (***Figure 7D and E***). For both comparisons, the correlation coefficient was higher when considering only X UP oogenesis genes. Moreover, as in *mes-4* M-Z- mutants, loss of LIN-15B caused *mes-3* M-Z- mutants to make a healthier germline (***Figure 5—figure supplement 3***). We conclude that MES-4 and PRC2 cooperate to ensure germline survival in M-Z- mutant larvae by repressing similar sets of X genes, especially sets of X-linked oogenesis genes, and that both operate through LIN-15B.

---

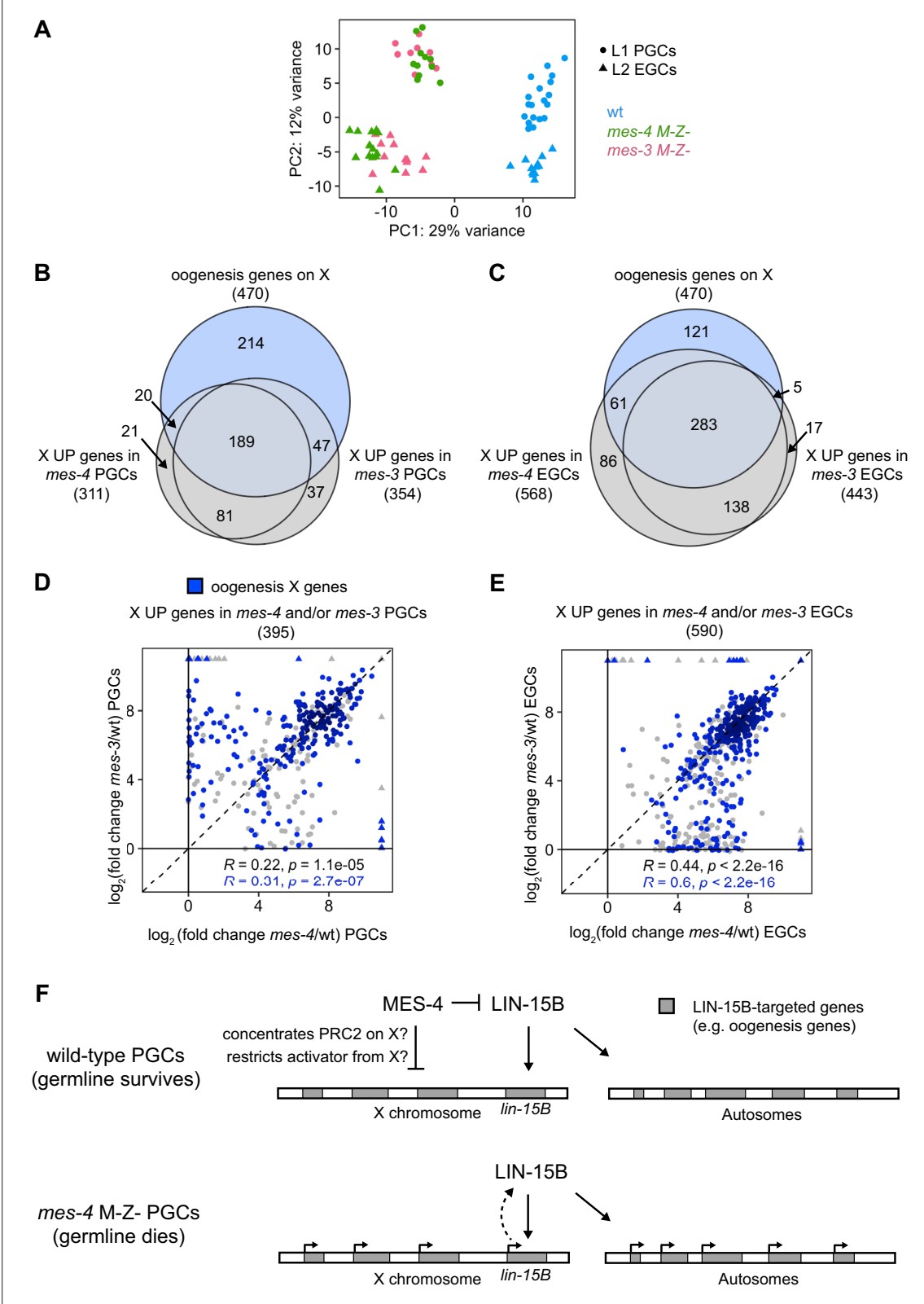

**Figure 7.** *mes-4* M-Z- and *mes-3* M-Z- nascent germlines misexpress a highly similar set of X genes, many of which are oogenesis genes. (**A**) Principal component analysis (PCA) including all replicates of wild-type (wt) (blue), *mes-4* M-Z- (green), and *mes-3* M-Z- (pink) primordial germ cells (PGCs) (circles) and early germ cells (EGCs) (triangles). The percentages of total variance across all samples described by the top two principal components are indicated. (**B, C**) Venn diagrams comparing X oogenesis genes to (**B**) X UP genes in *mes-4* and *mes-3* PGCs and to (**C**) X UP genes in *mes-4* and *mes-3*

*Figure 7 continued on next page*

*Figure 7 continued*

EGCs. (**D, E**) Scatterplots comparing log$_2$(fold change) (mutant vs. wt) of transcript abundance for X UP genes (gray circles) and X UP oogenesis genes (blue circles) in (**D**) *mes-4* and/or *mes-3* PGCs and in (**E**) *mes-4* and/or *mes-3* EGCs. The Spearman's correlation coefficients along with their p-values are indicated at the bottom of each scatterplot. (**F**) Cartoon model illustrating how MES-4 may protect germline survival by antagonizing LIN-15B. Our finding that the dose and gamete source of the X impact germline health upon loss of MES-4 focused our analysis on X genes. We identified LIN-15B as a key player in activating X genes, including *lin-15B*, and causing germline death upon loss of MES-4. LIN-15B may activate X genes directly by binding to those genes or indirectly by regulating one or more other transcription factors. MES-4 may repress *lin-15B* and other X genes by concentrating a repressor (e.g., PRC2) on the X or by restricting an activator (e.g., a histone acetyltransferase or LIN-15B) from the X. We do not know whether upregulation of X genes other than *lin-15B* and/or of autosomal genes contribute to germline death upon loss of MES-4.

The online version of this article includes the following figure supplement(s) for figure 7:

**Figure supplement 1.** Comparison of X and oogenesis gene misexpression in primordial germ cells (PGCs) and early germ cells (EGCs) dissected from various chromatin regulator mutants.

**Figure supplement 2.** Comparison of autosomal UP genes in *mes-4* and *mes-3* primordial germ cells (PGCs) and early germ cells (EGCs).

To improve our understanding of H3K36me3 marking and X repression, we extended our transcriptome analysis to PGCs from other H3K36me3 chromatin regulator mutants. The chromodomain protein MRG-1 is a candidate reader and effector of H3K36me3, and like MES-4 and PRC2, promotes germline development by repressing X genes (*Fujita et al., 2002*; *Takasaki et al., 2007*). To test whether MRG-1 represses the same set of X genes as MES-4 and PRC2, we profiled transcripts from PGCs hand-dissected from *mrg-1* M-Z- L1 larvae. We found that *mrg-1* PGCs misexpressed 440 X genes, 225 of which were also misexpressed in *mes-4* and *mes-3* PGCs and/or EGCs (*Figure 7—figure supplement 1*). We also profiled transcripts from PGCs hand-dissected from *met-1 M-Z-* L1 larvae, which lack the H3K36 HMT MET-1. We identified only eight upregulated X genes, which may explain why most *met-1* worms are fertile when reared at normal growth temperature (20°C). Our extended transcriptome analysis adds MRG-1 to the team of maternal regulators that ensure PGC survival and health by repressing the X.

## Discussion

In this study, we investigated how a maternally supplied chromatin regulator protects germline survival and promotes germline health. We found that nascent *C. elegans* germlines (PGCs and EGCs) that completely lack maternal MES-4 misexpress over 1000 genes, most of which are on the X chromosome. We further demonstrated that X misexpression is the cause of germline death in *mes-4* M-Z- mutants. Removal of a single transcription factor, LIN-15B, reduced X misexpression in the germline of *mes-4* mutant mothers (*mes-4* M+Z-) and was sufficient to allow most of their offspring (*mes-4* M-Z-) to develop full-sized germlines. Intriguingly, *lin-15B* is itself X-linked and misexpressed in nascent germlines that lack MES-4, highlighting *lin-15B* as a key target for MES-4 repression. We favor a model where maternal MES-4 promotes offspring germline development by preventing LIN-15B from activating a germline-toxic program of gene expression from the X chromosome, a program that is normally activated during oogenesis in hermaphrodites (*Figure 7F*). This work underscores how maternally supplied factors can guide development of specific tissues in offspring by protecting their transcriptome.

### A new strategy to isolate small numbers of cells for transcript profiling

We developed a hand-dissection strategy to isolate in under 30 min pure samples of PGCs and EGCs from synchronous *mes-4* M-Z- mutants that completely lack MES-4 for transcript profiling. We identified PGCs and EGCs during dissection using the highly and specifically expressed germline reporter GLH-1::GFP. This strategy offers numerous advantages over the popular cell isolation strategy, fluorescence-activated cell sorting (FACS), because FACS often (1) does not yield pure samples, (2) takes several hours to perform, during which time sample integrity may decline, and (3) requires large numbers of worms, which may be hard to obtain for some mutants, like *mes-4* mutants. Due to the high variation of transcript abundance measurements between biological replicates generated from ultra-low quantities of RNA (e.g., single pairs of PGCs and single sets of two EGCs), we acquired and analyzed a high number of biological replicates (at least 11) for each genotype. Even so, our differential expression analyses between PGCs or EGCs still had limited power to detect misregulated genes.

This encouraged us to use smFISH as an independent test of differential expression. Our smFISH findings matched our transcriptome findings for all genes tested, including two germline genes that were found to be expressed normally in *mes-4* PGCs, confirming that MES-4 does not regulate the expression of some germline genes. In sum, our hand-dissection strategy enabled us to significantly advance our understanding of MES-4's role in the germline. We envision that it may be a useful method for other *C. elegans* transcriptomics projects.

## MES-4's role in regulating germline gene expression

Our goal was to determine how complete loss of MES-4 impacts the transcriptome of PGCs as they begin to launch their germline program and before the germline exhibits declining health and death (*Capowski et al., 1991*; *Garvin et al., 1998*). Therefore, we chose to profile PGCs from recently hatched *mes-4* M-Z- L1 larvae that were fed for 30 min. Because PGCs do not fully turn on their germline program until after L1 larvae have fed for ~1.5 hr (*Butuči et al., 2015*), we also profiled EGCs from *mes-4* M-Z- L2 larvae that had fed for 20 hr. Thus, our samples flanked the major wave of zygotic genome activation in germ cells and enabled us to determine how loss of MES-4 impacts the transcriptome of nascent germlines. Maternal MES-4 binds to ~4100 protein-coding genes in embryos, most of which were previously expressed in the maternal germline and need to be expressed in offspring germlines (*Rechtsteiner et al., 2010*). Yet surprisingly, our transcriptome analysis found that lack of MES-4 did not impact the expression of many germline genes in PGCs and EGCs. Furthermore, our genetic findings show that *mes-4* mutants can develop a full-sized germline if they inherit X chromosomes that have a history of repression, demonstrating that MES-4 is not required for PGCs to launch a germline program.

Interestingly, MES-4 is required for the misexpression of germline genes in somatic tissues of several mutants, such as *spr-5; met-2*, *lin-15B*, and mutants of DREAM complex components (*Wang et al., 2005*; *Petrella et al., 2011*; *Carpenter et al., 2021*). This suggests that maternal MES-4 has tissue-dependent roles in gene regulation. Such context-dependent roles may be mediated by different H3K36me3 'reader' complexes, as observed in other organisms (*Yochum and Ayer, 2002*; *Cai et al., 2003*; *Chen et al., 2009*).

## Epigenetic inheritance of H3K36me3 patterns from parents is not required in offspring

There has been a concerted effort in recent years to determine how epigenetic inheritance impacts offspring health (e.g., *Heard and Martienssen, 2014*; *Klosin et al., 2017*; *Tabuchi et al., 2018*; *Kaneshiro et al., 2019*; *Fitz-James and Cavalli, 2022*). Maternal MES-4's role in propagating gamete-inherited H3K36me3 patterns through embryogenesis is a clear example of epigenetic inheritance (*Rechtsteiner et al., 2010*; *Kreher et al., 2018*). By taking advantage of the *gpr-1* genetic tool, we demonstrated that inheritance of H3K36me3 patterns from parents is not required for offspring germline development.

Although the germline can survive without having inherited H3K36me3 marking from parents, it cannot tolerate additional loss of either maternal MES-4 or maternal MET-1 (the two H3K36 HMTs in *C. elegans*). We speculate that maternal MES-4 and maternal MET-1 perform cooperative, but distinct, roles to restore sufficient levels and proper patterns of H3K36me3 marking to chromosomes inherited lacking H3K36me3, to allow germline development. In this scenario, we envision that maternal MET-1 first catalyzes new H3K36me3 marking on genes co-transcriptionally during the first wave of zygotic genome activation in embryonic PGCs, after which maternal MES-4 maintains MET-1-generated patterns of H3K36me3 through early germline development (*Furuhashi et al., 2010*; *Kreher et al., 2018*) to repress X genes and prevent germline death. De novo H3K36me3 marking on autosomes likely indirectly causes X repression in PGCs since most X genes are not transcribed in PGCs and therefore do not get de novo marked with H3K36me3. The importance of H3K36me3 marking is also highlighted by our finding that loss of the candidate H3K36me3 'reader' MRG-1 (homolog of yeast Eaf-3, fly MSL3, and human MRG15) (*Gorman et al., 1995*; *Eisen et al., 2001*; *Cai et al., 2003*; *Bertram and Pereira-Smith, 2001*; *Joshi and Struhl, 2005*) causes PGCs to misexpress a set of X genes similar to that caused by loss MES-4 and also causes death of the nascent germline. Our findings reveal the importance of H3K36me3 marking during early germline development, although that marking does not need to be inherited from parents.

## The distinct roles of MES-4 and MET-1 in germline development

Unlike MES-4, MET-1 does not have an essential role in germline development (*Andersen and Horvitz, 2007*; *Kreher et al., 2018*). *met-1* M-Z- worms are fertile when reared at standard 20°C growth temperature (*Andersen and Horvitz, 2007*), and *met-1* M-Z- PGCs misexpress very few genes (this study). One explanation is that in *met-1* mutant embryos maternal MES-4 can propagate patterns of pre-existing H3K36me3 inherited from parents to PGCs. This predicts that maternal MET-1's role in de novo H3K36me3 marking in response to transcriptional turn-on is not required so long as genes were pre-marked with H3K36me3 in parents. If so, maternal MET-1's role would be essential only if PGCs did not inherit pre-existing H3K36me3. Indeed, we found that maternal MET-1 was essential for germline development when PGCs inherited chromosomes lacking pre-existing H3K36me3 marking for maternal MES-4 to propagate. This scenario raises the fascinating possibility that in the absence of MET-1 function, maternal MES-4 may be able to faithfully transmit pre-existing patterns of H3K36me3 across generations and through development indefinitely.

## MES-4 likely represses X genes indirectly by catalyzing H3K36me3 on autosomes

Since MES-4 binding and its HMT activity are very low across almost the entire X chromosome (*Recht-steiner et al., 2010*), it is likely that MES-4 regulates expression of X genes indirectly in PGCs. One possible mechanism for indirect regulation is that MES-4 generates H3K36me3 on autosomes to repel and concentrate a transcriptional repressor on the X chromosome(s). An attractive candidate repressor is the H3K27 HMT PRC2: in germlines, PRC2 activity is concentrated on the X chromosome(s) (*Bender et al., 2004*), PRC2 represses a highly similar set of X genes as MES-4 (*Gaydos et al., 2012*; *Lee et al., 2017*; this study), and loss of PRC2 causes maternal-effect sterility, like loss of MES-4 (*Capowski et al., 1991*). H3K36me3's role in antagonizing methylation of H3K27 is well documented (*Schmitges et al., 2011*; *Yuan et al., 2011*; *Gaydos et al., 2012*; *Evans et al., 2016*). An alternative possibility is that H3K36me3 in germlines sequesters a transcriptional activator on autosomes and away from the X chromosome(s) as it does to the histone acetyltransferase (HAT) CBP-1 in *C. elegans* intestinal cells (*Cabianca et al., 2019*; *Georgescu et al., 2020*).

## The X-linked transcription factor LIN-15B is a major cause of germline death in *mes-4* mutants

We identified the THAP transcription factor LIN-15B as a cause of X misexpression in fertile germlines that lack MES-4 (*mes-4* M+Z- mutant mothers) and a major driver of germline death in their *mes-4* M-Z- mutant offspring. Interestingly, *lin-15B* is itself an X-linked gene that is upregulated in *mes-4* M-Z- mutant PGCs and EGCs. This suggests that *lin-15B* is a critical target of MES-4 repression of the X, to allow germline survival. There are likely additional targets as removal of LIN-15B does not allow all *mes-4* M-Z- mutants to make full germlines. We tested whether RNAi depletion of 20 candidate transcription factors and HATs improves germline health in *mes-4* mutants; we found no hits other than LIN-15B. Recent studies found that upregulation of *lin-15B* also causes sterility in *nanos* mutants and *set-2* (H3K4 HMT) mutants and leads to upregulation of other X genes (*Lee et al., 2017*; *Robert et al., 2020*). Those findings coupled with ours suggest that excessive LIN-15B activity causes germline-toxic levels of X-chromosome expression and that the germline uses multiple protective mechanisms to antagonize LIN-15B.

How LIN-15B causes X misexpression in germlines that lack MES-4 is unclear. Several studies have focused on LIN-15B's role as a repressor of germline genes in somatic tissues (*Wang et al., 2005*; *Petrella et al., 2011*; *Wu et al., 2012*). Recently, LIN-15B was shown to promote repressive H3K9me2 marking in the promoter of germline-specific genes in somatic cells (*Rechtsteiner et al., 2019*). In germlines, LIN-15B may activate expression of X genes indirectly, for example, by downregulating or antagonizing a repressor of X genes. Alternatively, LIN-15B may have context-dependent roles in gene expression, a well-known feature of many transcription factors (*Fry and Farnham, 1999*), and may directly activate expression of X genes. In support of this model, our analysis of LIN-15B ChIP data from whole embryos and larvae found that LIN-15B is associated with the promoter of many X genes that are upregulated in *mes-4* nascent germlines. Clarification of LIN-15B's mode of action in germlines requires analysis of patterns of LIN-15B binding in germlines, biochemical experiments, and identification of LIN-15B's functional partners.

## MES-4 and PRC2 regulation of X gene expression in nascent germlines

Since the focus of MES-4 and PRC2 regulation in nascent germlines is X repression, we wondered whether MES-4 and PRC2 (1) perform the germline's version of X-dosage compensation or (2) silence an oogenesis program, which is enriched for X-linked genes (*Ortiz et al., 2014*). The goal of X-dosage compensation is to equalize X gene expression in animals with one X (males) vs. two Xs (hermaphrodites). The mechanism of X-dosage compensation in somatic cells of worms is well understood (reviewed in *Meyer, 2022*). The mechanism in germ cells is currently not understood but is distinct from the mechanism used in somatic cells (reviewed in *Strome et al., 2014*). X-dosage compensation typically acts in either the 1X or 2X sex. Notably, the fertility of *mes-4* mutant worms is less dependent on the absolute number of X chromosomes in the germline than on whether the X(s) was inherited from the oocyte or sperm: 1X germlines with an $X^{oo}$ are typically sterile while 1X germlines with an $X^{sp}$ are typically fertile, and 2X germlines with one or two $X^{oo}$ are sterile while 2X germlines with two $X^{sp}$ can be fertile. Instead of MES-4 and PRC2 performing X-dosage compensation, we favor the view that they prevent misexpression of an oogenesis program, which is enriched for genes on the X (*Ortiz et al., 2014*). First, 60% of X genes upregulated in mutant PGCs and/or EGCs are oogenesis genes. Second, LIN-15B, a key X target of MES-4 and PRC2 repression, is an oogenesis gene and binds to the promoters of many other X-linked oogenesis genes. Third, upregulation of LIN-15B in *nanos* mutant germlines causes misexpression of oogenesis genes and death of nascent germ cells (*Lee et al., 2017*). Since the oocyte-inherited X chromosome has a history of expression of oogenesis genes, it may be prone to turning those genes on in, and thereby causing death of, nascent germ cells in offspring that lack MES-4 or PRC2. In support of this, offspring germ cells that lack MES-4 or PRC2 can develop into full-sized germlines if they inherit only X chromosomes from sperm, which were not turned on for oogenesis and have a history of repression. Together, our findings support a model where MES-4 and PRC2 protect germline survival and proliferation by silencing an X-linked oogenesis program that may interfere with the developmental fate of nascent germ cells.

## If MES-4 does not launch a germline program, what does?

An outstanding question is what launches the germline program in *C. elegans* PGCs (*Strome and Updike, 2015*). MES-4 has been a prime candidate since it transmits H3K36me marking of germline genes from parent germ cells to offspring germ cells and so has been invoked as passing a 'memory of germline' across generations. Our findings argue against MES-4 being essential to launch a germline program since *mes-4* mutant PGCs can undergo normal germline development if they inherit X chromosomes with a history of repression. However, MES-4 does promote the expression of at least some germline genes in PGCs and their descendants. Other attractive contenders for specifying the germline fate of PGCs have been germ granules and small RNAs. Several studies demonstrated that *C. elegans* germ granules (aka P granules) protect germline fate but are not needed to specify it (*Gallo et al., 2010*; *Updike et al., 2014*; *Knutson et al., 2017*). Among small RNAs, 22G small RNAs (22 bp long and starting with a G) are particularly attractive as possible germline determinants. They are bound by the argonaute CSR-1 and have been shown to target most germline-expressed genes and promote expression of some (*Claycomb et al., 2009*; *Conine et al., 2010*; *Wedeles and Claycomb, 2013*; *Cecere et al., 2014*). Complete loss of CSR-1 or DRH-3, an RNA helicase that generates 22G RNAs, causes sterility (*Duchaine et al., 2006*; *Claycomb et al., 2009*; *Gu et al., 2009*). However, hypomorphic mutations in the helicase domain of DRH-3 that abolish production of most 22G RNAs do not impact germline formation, suggesting that 22G RNAs are not needed to specify germline fate (*Gu et al., 2009*). We propose the intriguing possibility that in *C. elegans* germline fate is the default, which must be protected in the germline (e.g., by MES proteins and P granules) and opposed in somatic tissues (e.g., by DREAM and LIN-15B).

## Materials and methods
### Worm strains

All worms were maintained at 20°C on Nematode Growth Medium (NGM) plates spotted with *Escherichia coli* OP50 (*Brenner, 1974*). Strains generated (*) and used in this study are listed below. GLH-1 (Vasa) is a component of germ granules and is specifically and highly expressed in germline cells; *glh-1::GFP* was engineered into numerous strains to mark germ cells.

**DUP64** *glh-1(sams24[glh-1::GFP::3xFLAG]) I*
**SS1491\*** *glh-1(sams24[glh-1::GFP::3xFLAG]) I; mes-4(bn73)/tmC12[egl-9(tmIs1194)] V*
**SS1293\*** *glh-1(sams24[glh-1::GFP::3xFLAG]) I; mrg-1(qa6200)/qC1[dpy-19(e1259) glp-1(q339 )] III*
**SS1492\*** *mes-3(bn199)/tmC20 [unc-14(tmIs1219) dpy-5(tm9715)] glh-1(sams24[glh-1::GFP::3x FLAG]) I*
**SS1476\*** *met-1(bn200)/tmC20[unc-14(tmIs1219) dpy-5(tm9715)] glh-1(sams24[glh-1::GFP::3xF LAG]) I*
**SS1494\*** *met-1(bn200)/tmC20[unc-14(tmIs1219) dpy-5(tm9715)] glh-1(sams24[glh-1::GFP::3xF LAG]) I; mes-4(bn73)/tmC12[egl-9(tmIs1197)] V*
**SS1497\*** *glh-1(sams24[glh-1::GFP::3xFLAG]); oxTi421[eft-3p::mCherry::tbb-2 3'UTR+Cbr-unc-1 19(+)] X*
**SS1514\*** *glh-1(sams24[glh-1::GFP::3xFLAG]); mes-4(bn73)/tmC12[egl-9(tmIs1194)] V; oxTi421 [eft-3p::mCherry::tbb-2 3'UTR + Cbr-unc-119(+)] X*
**SS1503\*** *glh-1(sams24[glh-1::GFP::3xFLAG]) I; him-8(e1489) IV*
**SS1500\*** *glh-1(sams24[glh-1::GFP::3xFLAG]) I; him-8(e1489) IV; mes-4(bn73)/tmC12[egl-9(t mIs1194)] V*
**SS1498\*** *glh-1(sams24[glh-1::GFP::3xFLAG]) I; him-8(e1489) IV; oxTi421[eft-3p::mCherry::tbb-2 3'UTR+Cbr-unc-119(+)] X*
**SS1493\*** *glh-1(sams24[glh-1::GFP::3xFLAG]); him-8(e1489) IV; mes-4(bn73)/tmC12[egl-9(tmI s1194)] V; oxTi421[eft-3p::mCherry::tbb-2 3'UTR+Cbr-unc-119(+)] X*
**SS1515\*** *glh-1(sams24[glh-1::GFP::3xFLAG]) I; ccTi1594[mex-5p::GFP::gpr-1::smu-1 3'UTR+C br-unc-119(+), III: 680,195] III; hjSi20[myo-2p::mCherry::unc-54 3'UTR] IV*
**SS1516\*** *glh-1(sams24[glh-1::GFP::3xFLAG]) I; ccTi1594[mex-5p::GFP::gpr-1::smu-1 3'UTR+C br-unc-119(+), III: 680,195] III; hjSi20[myo-2p::mCherry::unc-54 3'UTR] IV; mes-4(bn73)/tmC1 2[egl-9(tmIs1194)] V*
**SS1517\*** *glh-1(sams24[glh-1::GFP::3xFLAG]) I; ccTi1594[mex-5p::GFP::gpr-1::smu-1 3'UTR+C br-unc-119(+), III: 680,195] III; hjSi20[myo-2p::mCherry::unc-54 3'UTR] IV; mes-4(bn73)/tmC1 2[egl-9(tmIs1194)] V; lin-15B(n744) X*
**SS1518\*** *met-1(bn200) glh-1(sams24[glh-1::GFP::3xFLAG]) I; ccTi1594[mex-5p::GFP::gpr-1:: smu-1 3'UTR+Cbr-unc-119(+), III: 680,195] III. hjSi20[myo-2p::mCherry::unc-54 3'UTR] IV*
**SS1511\*** *glh-1(sams24[glh-1::GFP::3xFLAG]) I; mes-4(bn73)/tmC12[egl-9(tmIs1194)] V; lin-15B( n744) X*
**SS1512\*** *mes-3(bn199)/tmC20[unc-14(tmIs1219) dpy-5(tm9715)] glh-1(sams24[glh-1::GFP::3xF LAG]) I; lin-15B(n744) X*

## Creation of *mes-3* and *met-1* null alleles by CRISPR-Cas9

The null alleles *mes-3(bn199)* and *met-1(bn200)* linked to *glh-1::GFP* were created by inserting TAAC TAACTAAAGATCT into the first exon of each locus. The resulting genomic edit introduced a TAA stop codon in each reading frame, a frame shift in the coding sequence, and a BglII restriction site (AGATCT) for genotyping. Alt-R crRNA oligos (IDT) were designed using CRISPOR (*Concordet and Haeussler, 2018*) and the UCSC Genome Browser (ce10) to produce highly efficient and specific Cas9 cleavage in the first exons of *mes-3* and *met-1*. Ultramer ssDNA oligos (IDT) containing 50 bp micro-homology arms were used as repair templates. A *dpy-10* co-CRISPR strategy (*Arribere et al., 2014*) was used to isolate strains carrying our desired mutations. Briefly, 2.0 µL of 100 µM *mes-3* or *met-1* crRNA and 0.5 µL of 100 µM *dpy-10* cRNA were annealed to 2.5 µL of 100 µM tracrRNA (IDT) by incubation at 95°C for 2 min, then at room temperature for 5 min, to produce sgRNAs. sgRNAs were complexed with 5 µL of 40 µM Cas9 protein at room temperature for 5 min, 1 µL of 40 µM *mes-3* or *met-1* repair template and 1 µL of 40 µM *dpy-10(cn64)* repair template were added, and the mix was centrifuged at 13,000 × *g* for 10 min. All RNA oligos were resuspended in duplex buffer (IDT, #11-05-01-03). Mixes were injected into one or both gonad arms of ~30 DUP64 adults. Transformant progeny were isolated and back-crossed 4× to DUP64.

## Isolation of single sets of two sister PGCs or two EGCs

L1 larvae hatched within a 30 min window in the absence of food were allowed to feed for 30 min to start PGC development. Larvae were partially immobilized in 15 µL drops of egg buffer (25 mM HEPES, pH 7.5, 118 mM NaCl, 48 mM KCl, 2 mM MgCl₂, 2 mM CaCl₂, adjusted to 340 mOsm) on poly-lysine-coated microscope slides and hand-dissected using 30-gauge needles to release their

gonad primordium (consisting of two connected sister PGCs and two somatic gonad precursors). Germline-specific expression of GLH-1::GFP was used to identify PGCs, which were separated from gonad precursor cells by mouth pipetting using pulled glass capillaries coated with Sigmacote (Sigma, #SL2) and 1% BSA in egg buffer. 7.5 mg/mL pronase (Sigma, #P8811) and 5 mM EDTA were added to reduce sticking of gonad primordia to the poly-lysine-coated slides and to weaken cell–cell interactions. Single sets of sister PGCs were transferred into 0.5 µL drops of egg buffer placed inside the caps of 0.5 mL low-bind tubes (USA Scientific, #1405-2600). Only PGCs that maintained bright fluorescence of GFP throughout isolation and were clearly separated from somatic gonad precursors were used for transcript profiling. Isolation of EGCs differed in three ways: (1) EGCs were dissected from L2 larvae that were fed for 20 hr after hatching, (2) the two EGCs that made up one sample may have come from different animals, and (3) the stage of each EGC could not be determined and therefore may have differed between samples. Tubes containing single sets of two sister PGCs or two EGCs were quickly centrifuged, flash frozen in liquid nitrogen, and stored at –80°C. A detailed protocol for isolating PGCs and EGCs from larvae is available upon request. At least 11 samples (biological replicates) of PGCs or EGCs were isolated for each condition.

## Isolation of RNAs from adult germlines

First day hermaphrodite and male adult worms (approximately 20–24 hr post-mid-L4 stage) were cut open with 30-gauge needles in egg buffer (see recipe above, except not adjusted to 340 mOsm) containing 0.1% Tween and 1 mM levamisole to extrude their gonads. Gonads were cut at the narrow 'bend' to separate the gonad region containing mitotic and early meiotic germ cells from the region that contains oocytes and/or sperm; the former was used for RNA profiling. 20–60 gonads were mouth pipetted into 500 µL Trizol reagent (Life Technologies, #15596018), flash-frozen in liquid nitrogen, and stored at –80°C for up to 1 month before RNA extraction.

To release RNAs from gonads in Trizol, gonads were freeze-thawed 3× using liquid nitrogen and a 37°C water bath, while vortexing vigorously between cycles. RNAs immersed in Trizol were added to phase-lock heavy gel tubes (Brinkmann Instruments, Inc, #955-15-404-5) and mixed with 100 µL of 1-bromo-3-chloropropane (BCP) (Sigma, #B9673), followed by room temperature incubation for 10 min. Samples were then centrifuged at 13,000 × $g$ and 4°C for 15 min to separate phases. RNAs in the aqueous phase were precipitated by mixing well with 0.7–0.8× volumes of ice-cold isopropanol and 1 µL 20 mg/mL glycogen, followed by incubation at –80°C for 1–2 hr and centrifugation at 13,000 × $g$ at 4°C for 30 min. RNA pellets were washed 3× with ice-cold 75% ethanol and then resuspended in 15 µL water. RNA concentration was determined with a Qubit fluorometer.

## Generation of cDNA sequencing libraries

### Primordial germ cells and early germ cells

Immediately after thawing PGCs and EGCs on ice, a 1:4,000,000 dilution of ERCC spike-in transcripts (Life Technologies, #4456740) was added to each sample. Double-stranded cDNAs from polyA(+) RNAs were generated using a SMART-seq method that combined parts of the Smart-seq2 (*Picelli et al., 2014*) and SMART-Seq v4 (Takara) protocols. Briefly, PGCs and EGCs were lysed at room temperature for 5 min in lysis buffer (Takara Bio, #635013) containing RNAse inhibitors (Takara Bio, #2313) to release mRNAs into solution. 1.2 µM custom DNA primer (IDT) was annealed to transcripts' polyA tails by incubating samples at 72°C for 3 min and then immediately placing them on ice. Reverse transcription to generate double-stranded cDNA was performed using 200 U SmartScribe, 1× first-strand buffer, 2 mM DTT (Takara Bio, #639537), 1 mM dNTPs (Takara Bio, #639125), 4 mM MgCl$_2$, 1 M betaine (Sigma, #B0300), 20 U RNAse Inhibitor (Takara Bio, #2313), and 1.2 µM custom template-switching oligo with a Locked Nucleic Acid analog (QIAGEN) at 42°C for 90 min, followed by 70°C for 15 min to heat-inactivate the reverse transcriptase. cDNAs were amplified using 20 cycles of PCR according to Takara's SMART-Seq v4 protocol using SeqAmp DNA Polymerase (Takara Bio, #638504) and a custom PCR primer. Amplified cDNAs were purified by SPRI using 1× Ampure XP beads (Agencourt, #A63881) and quantified using a Qubit fluorometer. All custom oligos contained a biotin group on their 5' end to ameliorate oligo concatemerization. Illumina's Nextera XT kit (Illumina, #FC-131-1096) was used with 350–400 pg cDNA as input to prepare dual-indexed Illumina RNA-sequencing libraries according to the manufacturer's instructions. Libraries were amplified using 14 cycles of PCR and purified by SPRI using 0.6× Ampure XP beads.

## Adult germlines

Illumina libraries were prepared from polyA(+) RNAs using the NEBNext Poly(A) mRNA Magnetic Isolation Module (NEB, #E7490) and the NEBNext Ultra RNA Library Prep Kit (NEB, #E7530) according to the manufacturer's instructions. 100 ng total RNA was used to generate cDNAs, which were then amplified using 15 cycles of PCR.

## Processing and analysis of RNA-seq data

### RNA sequencing

Library concentration was measured with a Qubit fluorometer, and fragment size was measured with a bioanalyzer or tapestation. Libraries were multiplexed and paired-end sequenced with 50 cycles on either an Illumina HiSeq2500 or NovaSeq 6000 SP flow cell at the QB3 Vincent J. Coates Genomics Sequencing Laboratory at UC Berkeley.

### Primordial germ cells and early germ cells

Paired-end Illumina sequencing reads were aligned to the ce10 (WS220) genome downloaded from Ensembl using Hisat2 (2.2.1). Samtools (1.10) was used to remove duplicate reads and reads with low mapping quality (MAPQ < 10) from the alignment file. The function featureCounts from the subread package (2.0.1) (*Liao et al., 2014*) was used to obtain gene-level read (fragment) counts using a ce10 (WS220) transcriptome annotation file (Ensembl), which additionally contained 92 ERCC spike-in transcripts. 11 low-quality transcript profiles that were likely caused by a failure to capture mRNAs from two PGCs or EGCs, a well-known problem in single-cell RNA-sequencing, were identified using the R package scuttle (1.2.0) (*McCarthy et al., 2017*). One replicate of *mrg-1* PGCs was removed from our analysis because its transcript profile contained reads that aligned to the deleted genomic sequence in the *mrg-1(qa6200)* allele and therefore was not dissected from a *mrg-1* M-Z- mutant larva. For details of the transcript profiles generated in this study, including which were filtered out of our analysis, see our NCBI GEO accession (GSE198552). The R package DESeq2 (1.32.0) (*Love et al., 2014*) was used to perform Wald tests that identified differentially expressed genes in mutant vs. wild-type samples. p-values were adjusted for multiple hypothesis testing by the Benjamini–Hochberg method to produce q-values. Genes with a q-value < 0.05 were considered differentially expressed. The scaling factors used by DESeq2 to normalize transcript profiles were calculated using the R package scran (1.20.1), which uses a pooling and deconvolution approach to deal with zero inflation in low-input RNA-seq data (*Lun et al., 2016*). The $\log_2$(fold change) values calculated by DESeq2 were shrunk using ashr in R (*Stephens, 2017*). Bigwig files containing normalized read coverage over the WS220/ce10 genome were generated with bamCoverage from deepTools (*Ramírez et al., 2016*) using library scaling factors computed by scran, a bin size of 5, and a smoothing window of 15. Average read coverage across biological replicates was computed using WiggleTools (*Zerbino et al., 2014*) and visualized on the UCSC Genome Browser. PCA was performed using DESeq2's plotPCA function and variance-stabilized counts from DESeq2's vst function. All visualizations of RNA-seq data were generated with the R packages ggplot2 (3.3.5) and ggpubr (0.4.0) or the UCSC Genome Browser. Gene set enrichment analyses were performed using hypergeometric tests in R and the DAVID Bioinformatics Resource 6.8 (*Huang et al., 2009*). The sizes of gene sets used for those tests are noted in the respective figure legends.

### Adult germlines

For transcript profiles from dissected adult germlines, data were processed as described above except the scaling normalization factors were computed by DESeq2.

## Gene sets

We restricted all of our analyses to protein-coding genes. The germline-specific and germline-enriched gene sets were previously defined in *Rechtsteiner et al., 2010* and *Reinke et al., 2004*, respectively. The MES-4-bound gene set was generated by analyzing MES-4 enrichment patterns on chromosomes using anti-FLAG ChIP-chip data from early embryo extracts that expressed a MES-4::GFP::FLAG transgene (*Rechtsteiner et al., 2010*). The soma-specific gene set was previously described in *Knutson et al., 2017* and refined in this study by removing genes that have >5 TPM in our RNA-seq data from wild-type dissected adult germlines. The oogenesis, spermatogenesis, and gender-neutral gene sets

were previously defined in *Ortiz et al., 2014*. Oogenesis genes are expressed at higher levels in dissected adult oogenic germlines than dissected adult spermatogenic germlines; spermatogenesis genes are the opposite. Gender-neutral genes are expressed at similar levels in both oogenic and spermatogenic germlines.

## smFISH in L1 larvae

100–200 gravid adult mothers were allowed to lay offspring in drops of S basal overnight. Starvation-synchronized L1 offspring were collected and fed HB101 bacteria in S basal for 5 hr. Fed L1s were washed 3–4 times with S basal to remove bacteria and then used for smFISH using the protocol described in *Ji and van Oudenaarden, 2012* with a few modifications. Briefly, L1s were fixed with 3.7% formaldehyde for 45 min at room temperature, followed by three washes with PBS-Tween (0.1%). Fixed L1s were incubated in 75% ethanol at 4°C overnight and up to 3 days. RNAs were hybridized to 25 nM RNA probe sets in hybridization buffer (2× SSC, 10% formamide, 0.1% Tween-20, and 0.1 g/mL dextran sulfate) at 37°C overnight. Afterward, larvae were washed 2× in hybridization buffer at 37°C for 30 min, the second of which included 1 ng/µL of DAPI. Larvae were washed 3× in PBS-Tween (0.1%) and mounted in anti-fade medium consisting of n-propyl gallate and Vectashield (Vector Laboratories, #H-1000). Mounted samples were immediately imaged on a spinning disk confocal microscope using a 100× oil objective to acquire 3D Z-stacks of PGCs; only Z-slices containing GLH-1::GFP signal were imaged. All RNA probe sets were conjugated to Quasar 670 fluorescent dye. We designed and purchased the *lsd-1*, *mbk-1*, *pek-1*, and *lin-15B* probe sets from Stellaris. The *cpg-2*, *pgl-3*, and *chs-1* probe sets were gifts from Dr. Erin Osborne Nishimura.

## Counting transcripts in 3D smFISH images

To batch process raw smFISH 3D images into transcript abundance measurements in PGCs, a custom pipeline written in Fiji (v2.1.0/1.53C) and MATLAB (R2020a) was developed. Much of the MATLAB code and strategy was adapted from *Raj et al., 2008* to create our pipeline. A Laplacian of Gaussian (LoG) filter was used to enhance the signal-to-noise contrast in smFISH images. Those filtered 3D images were thresholded by signal intensity to produce binary images. The 'imregionalmax' function from the Imaging Processing Toolbox in MATLAB was used to find and count regional maxima (transcript foci) in those binary images. To choose an appropriate signal intensity threshold for a 3D smFISH image, the number of detected regional maxima across 100 increasingly stringent thresholds applied to the image was plotted (*Raj et al., 2008*). In that plot, a range of thresholds that produced similar numbers of detected regional maxima was identified, and a threshold within that range was selected. Since threshold values were similar for all images within an image set (the collection of images acquired on the same day and for one probe set), an averaged threshold (across five images) was calculated and applied to all images in the set. Notably, threshold values were similar across different image sets. Manual counting of dots in a few smFISH images gave similar values as our semi-automated pipeline. To count regional maxima (transcripts) specifically in PGCs, we segmented PGCs in our smFISH images using Fiji. A 2D binary image mask of the PGCs was generated from a maximum-intensity Z-projection of the GLH-1::GFP image channel and applied to all Z-slices in the 3D smFISH images. The 2D binary image mask was made by first blurring the Z-projection using a large Gaussian kernel and then detecting edges in the blurred image. The number of PGC-specific transcripts was counted as regional maxima located within the segmented PGCs. Mann–Whitney tests were performed to compare transcript abundance between *mes-4* and wild-type PGCs. Numbers of analyzed PGCs per probe set per genotype (biological replicates) are indicated in *Figure 3C and F*. We excluded images of poor quality (blurry or very low DAPI or GLH-1::GFP) or with more than two PGCs prior to transcript analysis. No images were excluded from our analysis after transcript quantification.

## Germline size analysis

All analyses were performed by live imaging first day adults (approximately 20–24 hr post-mid-L4 stage) and evaluating germline size using the germline marker GLH-1::GFP. Adult germlines were classified into one of three categories: (1) 'full' if its size was similar to that of a wild-type adult germline, (2) 'partial' if it had at least ~15 GLH-1::GFP(+) germ cells but was not large enough to be classified as 'full', and (3) 'absent/tiny' if it had <15 GLH-1::GFP(+) germ cells. In rare ambiguous cases where a germline's size was intermediate between categories, the germline was assigned to the category of

the smaller size. To classify germline size for hermaphrodites, which have two gonad arms, only the size of the larger gonad arm was considered; in most cases, both gonad arms were similar in size. Two-tailed Fisher's exact tests were performed to test whether the proportion of worms with a full germline is higher or lower in one genotype vs. another genotype. Sample sizes of worms scored for each genotype are shown in the figures. Worms were excluded from the analysis if they were extremely sick.

Confocal microscopy was used to acquire live images of scored germlines in adults. Live adults were placed in a drop of 1 μL H$_2$O and 1 μL polystyrene microspheres (Polysciences, Inc, #00876) and then immobilized on 6% agarose pads. Images were acquired in Z-stacks using a 20× air objective and then converted to Z-projections of maximum intensity using Fiji. DIC projections were used to outline the body of worms.

## Gamete and progeny analysis

To determine whether *mes-4* M-Z- X$^{sp}$ males that made full-sized germlines also produced sperm, we imaged ethanol-fixed and DAPI-stained males using a spinning disk confocal microscope. Males were assigned to one of three bins by the number of sperm (small, dense DAPI-stained bodies in the proximal region of the germline) they contained: (1) 0–10 sperm, (2) 10–50 sperm, or (3) >50 sperm. To determine whether *mes-4* M-Z- X$^{sp}$X$^{sp}$ hermaphrodites that made full-sized germlines also produced progeny, live imaging was used to determine whether they contained at least one egg in their uterus, after which single egg-containing hermaphrodites were transferred to individual plates and scored for production of viable progeny. In this way, hermaphrodites were assigned to one of three bins: (1) 'no eggs', (2) 'eggs' (but did not produce viable progeny), and (3) 'eggs and progeny.' Only males and hermaphrodites with full-sized germlines were considered in the analysis. We used two-sided Fisher's exact tests to compare what proportion of wild-type and *mes-4* M-Z- mutant males contained >50 sperm and hermaphrodites contained 'eggs and progeny.'.

## Tracking X-chromosome inheritance patterns

Methods used to track X-chromosome inheritance are diagrammed in *Figure 4—figure supplement 1*. To identify F1 male offspring that received their single X from the oocyte (X$^{oo}$) or the sperm (X$^{sp}$), crosses were performed with one parent contributing an X-linked *eft-3p::mCherry* transgene. F1 male offspring that inherited the transgene were easily distinguished by bright cytoplasmic mCherry fluorescence in their soma. A *gpr-1* overexpression allele was used to generate F1 hermaphrodite offspring whose germline inherited either two genomes from the sperm or two genomes from the oocyte (*Besseling and Bringmann, 2016*; *Artiles et al., 2019*). Those non-Mendelian offspring were visually identified by patterns of mCherry fluorescence in their pharyngeal muscle cells (*myo-2p::mCherry* X) (*Figure 4—figure supplement 1*).

## RNA interference (RNAi) depletion of gene products

RNAi was performed by feeding worms *E. coli* HT115 bacteria that carry a gene target's DNA sequence in the L4440 vector (*Kamath et al., 2003*). Most RNAi constructs were obtained from the Ahringer RNAi library and sequence confirmed. *lin-15B*, *lsy-2*, *nfya-1*, *eor-1*, and *sma-9* RNAi constructs were generated for this study. RNAi constructs were streaked onto LB agar plates containing 100 μg/mL carbenicillin and 10 μg/mL tetracycline. Single clones were cultured overnight (14–17 hr) at 37°C in LB and carbenicillin (100 μg/mL). The following day, RNAi cultures were spotted onto 6 cm NGM plates containing 1 mM IPTG and 100 μg/mL carbenicillin (both added by top spreading), and then left to dry for 2 days at room temperature in the dark. To deplete both maternal load and embryo-synthesized gene product, we placed L4-stage larval mothers onto RNAi plates, grew them 1 day to reach adulthood, then moved those adults to new RNAi plates to lay offspring. RNAi was done at 20°C. Germline size was scored in first day adult offspring as described above.

## Spinning-disk confocal microscopy

All images were acquired using a spinning-disk confocal microscope equipped with a Yokogawa CSU-X1 confocal scanner unit, Nikon TE2000-E inverted stand, Hamamatsu ImageEM X2-CCD camera, solid state 405, 488, 561, 640 nm laser lines, 460/50, 525/50, 593/40, 700/75 nm (EM/BP) fluorescence filters, DIC, Nikon Plan Apo VC ×20/0.5 air objective, Nikon Plan Apo ×100/1.40 oil objective,

and Micro-Manager software (1.4.20). Image processing for images was done in Fiji (2.1.0/1.53C) and Photoshop.

## Transcription factor analyses

### Analyses of ChIP data

Bed files containing transcription factor (TF) binding sites (ChIP-chip or ChIP-seq 'peaks') across the ce10 genome were downloaded from the modENCODE (*Gerstein et al., 2010*) and modERN (*Kudron et al., 2018*) websites. Each bed file was loaded into R and converted into a GRanges object using the GenomicRanges package (*Lawrence et al., 2013*). TF binding sites were assigned to genes using the package ChIPSeeker (*Yu et al., 2015*). A TF binding site was assigned to a gene if it overlapped with a gene's TSS region (500 bp upstream of TSS). If a TF had more than one set of binding site data (e.g., two ChIP-seq experiments), each set was processed separately and then the TF-assigned genes were merged. TFs enriched in promoters of X-linked UP genes in *mes-4* PGCs and/or EGCs (584 genes) compared to all X-linked genes (2808) were identified using hypergeometric tests (p-value<0.05).

### Analyses of DNA motifs

Position-weight matrices for known *C. elegans* TF DNA motifs were downloaded from the CisBP database (*Weirauch et al., 2014*). X-linked genes that contain a TF's motif in their promoter (500 bp upstream of TSS) were identified using FIMO from the Meme suite in R using default parameters (*Grant et al., 2011*). Motifs significantly enriched in promoters of X-linked UP genes in *mes-4* PGCs or EGCs (584 genes) compared to all X-linked genes (2808) were identified using hypergeometric tests (p-value<0.05).

## Materials and data availability

All sequencing data generated in this study were deposited in NCBI GEO under accession GSE198552. Newly created materials and protocols are available upon request. Custom scripts used to process and analyze data are available in the following GitHub repository: https://github.com/ccockrum-ucsc/mes-transcriptomics, (copy archived at swh:1:rev:20c99ad1168e8b8705004dd4e979bf39a5443f60, *Cockrum and Strome, 2022*). *Supplementary file 1* contains differentially expressed genes for all comparisons we analyzed. *Supplementary file 2* contains a list of DNA/RNA oligos used in this study.

## Acknowledgements

We thank past and present members of the Strome lab and Grant Hartzog for helpful discussions, Ben Abrams for microscopy advice, and Joshua Arribere for help with statistical analyses. We thank Dustin Updike for sharing the *glh-1::GFP* strain and Erin Osborne Nishimura for providing RNA probe sets for smFISH. This work was supported by NIH grants T32GM008646 and F31GM125305 to CC and R01GM34059 to SS.

## Additional information

### Funding

| Funder | Grant reference number | Author |
| --- | --- | --- |
| National Institutes of Health | T32GM008646 | Chad Steven Cockrum |
| National Institutes of Health | F31GM125305 | Chad Steven Cockrum |
| National Institutes of Health | R01GM34059 | Susan Strome |

The funders had no role in study design, data collection and interpretation, or the decision to submit the work for publication.

## Author contributions
Chad Steven Cockrum, Conceptualization, Data curation, Software, Formal analysis, Funding acquisition, Investigation, Visualization, Methodology, Writing - original draft, Project administration, Writing - review and editing; Susan Strome, Conceptualization, Resources, Supervision, Funding acquisition, Writing - original draft, Writing - review and editing

## Author ORCIDs
Chad Steven Cockrum http://orcid.org/0000-0002-4080-9184
Susan Strome http://orcid.org/0000-0001-9496-7412

## Decision letter and Author response
Decision letter https://doi.org/10.7554/eLife.77951.sa1
Author response https://doi.org/10.7554/eLife.77951.sa2

# Additional files

## Supplementary files
• Supplementary file 1. List of differentially expressed genes in mutant vs. wild-type (wt) primordial germ cells (PGCs) and early germ cells (EGCs). Six tables show the results of differential expression analysis of *mes-4* vs. wt PGCs, *mes-4* vs. wt EGCs, *mes-3* vs. wt PGCs, *mes-3* vs. wt EGCs, *mrg-1* vs. wt PGCs, and *met-1* vs. wt PGCs. Numbers of biological replicates for each condition are in the legend of *Figure 7—figure supplement 1*. Each table includes the wormbase ID and public ID for each differentially expressed gene in the respective comparison, a column that indicates whether a differentially expressed gene was upregulated or downregulated, and a column that indicates whether each gene is X-linked or autosomal.

• Supplementary file 2. List of DNA/RNA oligos used in the study. For each oligo, its DNA or RNA sequence and a description of how it was used in the study are included.

• MDAR checklist

## Data availability
All sequencing data generated in this study were deposited in NCBI GEO under accession GSE198552.

The following dataset was generated:

| Author(s) | Year | Dataset title | Dataset URL | Database and Identifier |
|---|---|---|---|---|
| Cockrum C, Strome S | 2022 | Maternal H3K36 and H3K27 HMTs protect germline immortality via regulation of the transcription factor LIN-15B | https://www.ncbi.nlm.nih.gov/geo/query/acc.cgi?acc=GSE198552 | NCBI Gene Expression Omnibus, GSE198552 |

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
