## [Editor Report]

This study provides a compelling and significant advance on the understanding of how gene regulation by the histone methyltransferase MES-4 underlies germ cell survival in *C. elegans*, with the major claims being nicely substantiated. The critical and surprising finding is that the degeneration of *mes-4* mutant primordial germ cells is due to inappropriate upregulation of genes on the silenced X chromosome, and not a failure to activate germline-expressed genes, though reduced levels of germline gene expression were observed. An X-linked target of *mes-4*, *lin-15b*, is necessary for the degeneration phenotype. The claim is compellingly supported.

---

## [Decision Letter]

**Decision letter after peer review:**

Thank you for submitting your article "Maternal H3K36 and H3K27 HMTs protect germline immortality via regulation of the transcription factor LIN-15B" for consideration by *eLife*. Your article has been reviewed by 3 peer reviewers, including Yukiko M Yamashita as the Reviewing Editor and Reviewer #1, and the evaluation has been overseen by Marianne Bronner as the Senior Editor.

Essential revisions:

Overall, the study is well done and the major claims are supported by the data provided in the manuscript. Whereas the upregulation of X-linked genes as the major cause of PGC death is well supported, it is less well supported that mes-4 and the Polycomb complex genes are not needed for turn on of germline gene expression or repression of somatic gene expression. Indeed the data suggests that these genes are needed for appropriate germline gene turn on and soma gene repression as defects in turn on and repression are clear in EGCs. Germline genes appear to be expressed in mutants, but levels for a significant number were observed to be reduced in mutants. The lower level of defects observed in PGCs may be due to profiling before the zygotic genome is fully activated. One might argue that the number of genes found to be downregulated is much smaller than the number of genes normally expressed in the germline, suggesting that most genes turned on normally. However, profiling gene expression in a very small number of cells may only have enough power to detect downregulation in genes that have relatively high expression. The authors could determine this computationally. More simply, looking at the wild-type expression level of the genes downregulated in mutants would be informative – i.e., are they predominantly in the top brackets of germline gene expression? If so, the author could gain power to detect downregulation of lower expression genes by binning genes by expression level (e.g, 10 levels) and then comparing expression in wt and mutants. The current analyses in this paper are not sufficient to conclude that mes-4 and the Pc genes are not needed for turn on of germline gene expression and repression of somatic gene expression. A bit further digging into their data might provide such evidence. If these analyses are not possible, the authors can edit the text to weaken the conclusion surrounding mes-4/Pc being not required for turning on germline genes.

*Reviewer #2 (Recommendations for the authors):*

Line 358 – The text needs more explanation of how the gpr-1 "genetic tool" works. A sentence or two would suffice. In particular, it needs to be stated explicitly whether this tool can be used to change the chromatin state of all chromosomes or only the X, and if so why or why not. Perhaps this clarification would help with all the subsequent confusion described below.

Section starting at line 371 – This section is very hard to follow. It's not easy to understand the various conditions in which the X is being assessed, especially in the absence of scrutinizing Figure 3. Specifically, it is not clear why met-1 is mentioned here (line 381). Additionally, the sentence starting at line 386 only discusses the H3K36me3 patterns generally, but the resulting inference is that the critical role is X regulation. Perhaps this should be explicitly stated, as the way it's written now feels cut short.

Even more problematic, the subsequent paragraph (line 389) is where met-1 seems to be actually discussed, and it again took multiple readings to understand the ultimate conclusion, which I interpret to be that MES-4 does have a role in promoting germline gene expression, but because MET-1 is also needed to promote germline gene expression due to the division of labor between the two HMTs (de novo vs maintenance methylation), the loss of either is sufficient to impair activation of autosomal germline gene expression. The final sentence needs to mention the X explicitly or clarify what is going on regarding activation of autosomal genes as well. What is confusing is that the focus is initially on the X, but when the mes-4 and met-1 relationship is being studied in Figure 3b, it is not stated explicitly that the X does not have much H3K36me3 in wild type germ cells (at least until oogenesis is initiated in adults). So does the requirement for met-1 and mes-4 have to do with autosomal genes? Indeed, perhaps the model in figure 5 needs to specify the effect on autosomal genes and a role for MET-1 as well as for the X and MES-4 and LIN-15B. Even better would be to analyze gene expression in dissected EGCs from met- 1 and maybe also met-1;mes-4 mutants.

Line 488 – This sentence is hard to follow – needs re-writing.

*Reviewer #3 (Recommendations for the authors):*

Specific comments

1. The authors clearly demonstrated that a major defect of mes-4 mutants (and polycomb mutants) is the upregulation of X-linked gene expression, and that this contributes strongly to their sterility. It is striking that reducing X chromosome dosage in the mutants rescues germline development and fertility (though not fully). This raises an obvious question, not mentioned by the authors, as to whether the mes-4/Polycomb system plays a role in X chromosome dosage compensation in the germline. It would be helpful for the authors to discuss this possibility.

2. A second conclusion made by the authors is mes-4 mutant PGCs and EGCs have normal turn on of most germline genes and normal repression of somatic genes. This conclusion is not convincing and needs further support. The PGC profiling in L1s appears to have been done at a time when the zygotic genome is not fully activated, which would have limited the ability to assess turn-on of germline gene expression. The zygotic genome of PGCs is predominantly quiescent in embryos and undergoes major activation in L1 PGCs after feeding. The timing of full activation is not known, but Butuci et al. 2015 showed that RNAP ser2 phosphorylation, a marker of activation, has not reached a full level by 1 hour post-feeding. Here the authors collected L1 PGCs 30 minutes after feeding, a time when zygotic genome is apparently not fully activated. It is also possible that some fraction of the RNAs detected at this time point are not zygotic transcripts, but rather are maternally inherited RNAs associated with P granules. Consistent with the possibility that low zygotic transcription at 30' post-feeding L1 PGCs may prevented the ability to detect defects in turn on of gene expression, the authors' profiling of L2 EGCs found many more changes than in early L1s, and indeed in Figure 1 figure supplement 1 they show a significant enrichment of germline-enriched genes among those downregulated in mes-4 (panel G) and a weaker but significant enrichment of the (autosomal) soma-specific genes among the upregulated genes (panel I). On face value, these results are in line with a role for MES-4 in promoting germline gene expression and inhibiting soma gene expression in early germ cells, in addition to the demonstrated role in repressing X chromosome gene expression. The authors should address these issues.

a) The timing of PGC transcriptional turn on and consequent ability to robustly detect mRNAs in wild-type needs to be determined to fully interpret the PGC and EGC profiling experiments. If a gene hasn't strongly turned on yet in wt, then it wouldn't be able to be scored as being down regulated in mutants. To address this, the authors could assess germline expressed genes by FISH, comparing signals in late embryos, 30' fed L1s, and EGCs. In addition to germline-specific or germline-enriched genes, the timing of germline expressed genes with non-enriched expression should be assayed, which should include ubiquitously expressed genes.

b) The striking high X-linked gene expression in early L1 PGCs and the later observed downregulation of germline genes and upregulation of soma genes in L2 EGCs suggests the possibility that these two regulatory processes may be independent and occur at different times. That is, the upregulation in X-linked gene expression in PGCs might have occurred in embryos, before the major turn on of gene expression in fed L1s, and subsequently there may be a defect in the normal turn on of germline gene expression and repression of somatic gene expression in larvae. The authors should conduct FISH experiments to determine the timing of the different defects.

c) The germline and soma gene sets used in this paper are small in number and limit the strength of the conclusions and ability to conclude whether most germline genes turn on normally in mes-4 mutants. The rationale for only assessing a minority of germline expressed genes and for limiting assays to germline genes with germline-specific (n=168) or germline-enriched (n=2178) expression was not clear. To better assess whether PGCs and EGCs have normal turn on of most germline genes and normal repression of somatic genes, the authors should analyse more comprehensive gene lists. To address whether germline genes are turned on normally or not, the authors should assess most germline-expressed genes, including ubiquitously expressed genes. Germline expressed genes in L2 EGCs can be obtained from Cao et al. 2017, from adult germlines in Serizay et al., 2021 and other sources. Genes expressed only in soma can also be found in these and other publications. It should also be tested computationally whether the potential sparseness of the profiling data has limited the ability to observe downregulation of individual germline genes to those with relatively high expression in wild-type. To gain power, the authors could bin germline genes of different expression levels and ask whether expression of the group is lower in mutants.

3. The authors should include tables of all of the gene sets they identified to be up/down in PGCs and EGCs in different mutant backgrounds, i.e., up in mes-4 L1 PGCs, down in mes-4 L1 PGCs,.…

4. The GEO submission indicates the availability of a file of normalized counts that I couldn't find: "A matrix of normalized counts for samples of PGCs and EGCs is included in this submission: pgc_egcs_countsNormalized.txt pgc_egcs_countsNormalized.txt"

---

## [Author Response]

Essential revisions:

Point (#1) We moved analysis of *mes-4* mutant EGCs (Early Germ Cells from L2 larvae) from supplement to Figure 2 of the main paper for comparison with *mes-4* mutant PGCs (from recently hatched L1s). And we addressed the concern that germ cells may not fully display transcriptional defects in mutant worms until after wild-type worms fully turn on the germline transcription program (after L1 larvae have fed for ~1.5 hours). PGCs and EGCs flank that major wave of turn-on of germline genes, and both show transcriptional defects in *mes-4* mutant larvae.

Point (#2) We pointed out and discussed that EGCs display all 3 of the tested categories of gene mis-regulation: reduced expression of some germline genes, inappropriate expression of some somatic genes, and inappropriate expression of genes on the X chromosome. Inappropriate expression of genes on the X is certainly the most dramatic defect in *mes-4* mutant PGCs and EGCs and is the cause of germline death. Importantly, rescue of germline health by modulating the X-chromosome composition in mutant worms demonstrates that the germline gene expression program in *mes-4* mutants is “normal enough” to support germline development.

Point (#3) We clarified how we used the *gpr-1* tool to generate XX hermaphrodite germlines that inherited both X chromosomes (with a history of repression) and 10 autosomes from the sperm.

Point (#4) We better explained the likely roles of MES-4 and MET-1 (the other H3K36 HMT in worms) in rescuing the health of *mes-4* mutant germlines that inherit chromosomes lacking H3K36me3.

Point (#5) We included in Discussion a section on whether up-regulation of X genes in *mes-4* mutant PGCs and EGCs reflects a defect in X-chromosome dosage compensation or a defect in keeping the oogenesis program (which is enriched for X-linked genes) quiet in the nascent germline. We favor the latter, based on new analyses in Results.

Overall, the study is well done and the major claims are supported by the data provided in the manuscript. Whereas the upregulation of X-linked genes as the major cause of PGC death is well supported, it is less well supported that mes-4 and the Polycomb complex genes are not needed for turn on of germline gene expression or repression of somatic gene expression. Indeed the data suggests that these genes are needed for appropriate germline gene turn on and soma gene repression as defects in turn on and repression are clear in EGCs. Germline genes appear to be expressed in mutants, but levels for a significant number were observed to be reduced in mutants. The lower level of defects observed in PGCs may be due to profiling before the zygotic genome is fully activated. One might argue that the number of genes found to be downregulated is much smaller than the number of genes normally expressed in the germline, suggesting that most genes turned on normally. However, profiling gene expression in a very small number of cells may only have enough power to detect downregulation in genes that have relatively high expression. The authors could determine this computationally. More simply, looking at the wild-type expression level of the genes downregulated in mutants would be informative – i.e., are they predominantly in the top brackets of germline gene expression? If so, the author could gain power to detect downregulation of lower expression genes by binning genes by expression level (e.g, 10 levels) and then comparing expression in wt and mutants. The current analyses in this paper are not sufficient to conclude that mes-4 and the Pc genes are not needed for turn on of germline gene expression and repression of somatic gene expression. A bit further digging into their data might provide such evidence. If these analyses are not possible, the authors can edit the text to weaken the conclusion surrounding mes-4/Pc being not required for turning on germline genes.

As suggested, we binned genes by expression level in wild-type germlines into deciles and analyzed gene expression levels between *mes-4* mutant and wild-type PGCs or EGCs in each decile. Indeed, downregulation was most pronounced in the top decides, but was still apparent in the bottom deciles (see Figure 2—figure supplement 2). We also mentioned in Discussion our limited power to detect gene mis-regulation using RNA-seq of pairs of cells.

We changed our language in Results from “turn-on of genes” to “expression of genes” and ”transcript levels from genes,” as a more accurate description of what our experiments measured. And as noted in Point #2 above, we acknowledged and discussed that EGCs display all 3 of the tested categories of gene mis-regulation: reduced expression of some germline genes, inappropriate expression of some somatic genes, and inappropriate expression of genes on the X chromosome (see Figure 2). Inappropriate expression of genes on the X is certainly the most dramatic defect in *mes-4* mutant PGCs and EGCs and is the cause of germline death. Importantly, rescue of germline health by modulating the X chromosome composition in mutant worms demonstrates that the germline gene expression program in *mes-4* mutants is “normal enough” to support germline development.

Reviewer #2 (Recommendations for the authors):Line 358 – The text needs more explanation of how the gpr-1 "genetic tool" works. A sentence or two would suffice. In particular, it needs to be stated explicitly whether this tool can be used to change the chromatin state of all chromosomes or only the X, and if so why or why not. Perhaps this clarification would help with all the subsequent confusion described below.

As noted in Point #3 above, we clarified how we used the *gpr-1* tool to generate XX hermaphrodite germlines that inherited both X chromosomes (with a history of repression) and 10 autosomes from the sperm.

Section starting at line 371 – This section is very hard to follow. It's not easy to understand the various conditions in which the X is being assessed, especially in the absence of scrutinizing Figure 3. Specifically, it is not clear why met-1 is mentioned here (line 381). Additionally, the sentence starting at line 386 only discusses the H3K36me3 patterns generally, but the resulting inference is that the critical role is X regulation. Perhaps this should be explicitly stated, as the way it's written now feels cut short.Even more problematic, the subsequent paragraph (line 389) is where met-1 seems to be actually discussed, and it again took multiple readings to understand the ultimate conclusion, which I interpret to be that MES-4 does have a role in promoting germline gene expression, but because MET-1 is also needed to promote germline gene expression due to the division of labor between the two HMTs (de novo vs maintenance methylation), the loss of either is sufficient to impair activation of autosomal germline gene expression. The final sentence needs to mention the X explicitly or clarify what is going on regarding activation of autosomal genes as well. What is confusing is that the focus is initially on the X, but when the mes-4 and met-1 relationship is being studied in Figure 3b, it is not stated explicitly that the X does not have much H3K36me3 in wild type germ cells (at least until oogenesis is initiated in adults). So does the requirement for met-1 and mes-4 have to do with autosomal genes? Indeed, perhaps the model in figure 5 needs to specify the effect on autosomal genes and a role for MET-1 as well as for the X and MES-4 and LIN-15B. Even better would be to analyze gene expression in dissected EGCs from met- 1 and maybe also met-1;mes-4 mutants.

Yes, the MET-1 part of the story was surprising to us and is complicated. We changed the wording in that section to be easier for readers to follow and added a new section to Discussion. Although MET-1 is not needed to generate a fertile germline, we thought it important to include it in our analysis of germlines that inherit H3K36me3(-) chromosomes, since MET-1 is the only other known H3K36 HMT in worms besides MES-4. As noted in Point #5 above, we discussed in more detail in our revised paper the relative roles of MES-4 and MET-1 in embryos: in embryos MET-1 can generate H3K36me3 de novo; MES-4 maintains H3K36me3 but cannot generate it de novo (Furuhashi et al., 2010). We speculate that in PGCs that inherit H3K36me3(-) chromosomes, when some transcription is initiated during embryogenesis (e.g. Kawasaki et al., 1998; Subramaniam and Seydoux, 1999; Spencer et al., 2011), MET-1 de novo generates H3K36me3 in a transcription-coupled manner, and MES-4 maintains new H3K36me3. Our results show that maternal provision of both activities is needed to rescue the health of *met-1; mes-4* mutant germlines that inherit H3K36me3(-) chromosomes. As mentioned above, we think that the important take-aways are that inherited H3K36me3 marking is not critical for germline development and that re-establishment of marking is important and requires both enzymes. Re-established marking may resemble the marking that is usually inherited and maintained by MES-4. Our model (Figure 7F) already discusses 2 indirect models of how H3K36me3 marking on autosomes may impact transcription of genes on the X. We modified our model panel to focus on oogenesis genes and to include those on the autosomes as well as on the X.

Line 488 – This sentence is hard to follow – needs re-writing.

Done.

Reviewer #3 (Recommendations for the authors):Specific comments1. The authors clearly demonstrated that a major defect of mes-4 mutants (and polycomb mutants) is the upregulation of X-linked gene expression, and that this contributes strongly to their sterility. It is striking that reducing X chromosome dosage in the mutants rescues germline development and fertility (though not fully). This raises an obvious question, not mentioned by the authors, as to whether the mes-4/Polycomb system plays a role in X chromosome dosage compensation in the germline. It would be helpful for the authors to discuss this possibility.

See Point #5 above and our expanded response to Reviewer #1.

2. A second conclusion made by the authors is mes-4 mutant PGCs and EGCs have normal turn on of most germline genes and normal repression of somatic genes. This conclusion is not convincing and needs further support. The PGC profiling in L1s appears to have been done at a time when the zygotic genome is not fully activated, which would have limited the ability to assess turn-on of germline gene expression. The zygotic genome of PGCs is predominantly quiescent in embryos and undergoes major activation in L1 PGCs after feeding. The timing of full activation is not known, but Butuci et al. 2015 showed that RNAP ser2 phosphorylation, a marker of activation, has not reached a full level by 1 hour post-feeding. Here the authors collected L1 PGCs 30 minutes after feeding, a time when zygotic genome is apparently not fully activated. It is also possible that some fraction of the RNAs detected at this time point are not zygotic transcripts, but rather are maternally inherited RNAs associated with P granules. Consistent with the possibility that low zygotic transcription at 30' post-feeding L1 PGCs may prevented the ability to detect defects in turn on of gene expression, the authors' profiling of L2 EGCs found many more changes than in early L1s, and indeed in Figure 1 figure supplement 1 they show a significant enrichment of germline-enriched genes among those downregulated in mes-4 (panel G) and a weaker but significant enrichment of the (autosomal) soma-specific genes among the upregulated genes (panel I). On face value, these results are in line with a role for MES-4 in promoting germline gene expression and inhibiting soma gene expression in early germ cells, in addition to the demonstrated role in repressing X chromosome gene expression. The authors should address these issues.

We have now addressed a role for MES-4 in promoting germline gene expression and inhibiting somatic gene expression. See Point #2 above and our expanded response to the Editor’s Essential Revisions.

a) The timing of PGC transcriptional turn on and consequent ability to robustly detect mRNAs in wild-type needs to be determined to fully interpret the PGC and EGC profiling experiments. If a gene hasn't strongly turned on yet in wt, then it wouldn't be able to be scored as being down regulated in mutants. To address this, the authors could assess germline expressed genes by FISH, comparing signals in late embryos, 30' fed L1s, and EGCs. In addition to germline-specific or germline-enriched genes, the timing of germline expressed genes with non-enriched expression should be assayed, which should include ubiquitously expressed genes.

The major wave of zygotic genome activation (ZGA) in germ cells commences after L1s hatch and feed (Schaner et al., 2003). Based on staining PGCs for elongating RNA Pol II, the exact timing of ZGA is probably by 1.5 hours of feeding and before the PGCs start proliferating (Schaner et al., 2003; Butuci et al., 2015). Before that major wave of ZGA, numerous papers have demonstrated that some transcription commences in PGCs during embryogenesis (e.g. Kawasaki et al., 1998; Subramaniam et al., 1999; Furuhashi et al., 2010; Spencer et al., 2011). We profiled PGCs isolated from 30’ fed L1s within 30-60 min after hatching and EGCs isolated from L2s after ~20 hrs of feeding. Our PGC samples likely precede the major wave of ZGA in germ cells, while EGCs likely follow it, so we essentially did the suggested “flank the ZGA” experiment. We observed upregulation of X-linked gene expression in both PGCs and EGCs from mutants (see Figures 1 and 2). We also observed that EGCs displayed reduced expression of some germline genes and increased expression of some somatic genes, which we now point out and discuss in the paper, as noted in Point #2 above and our expanded response to the Editor’s Essential Revisions (see Figure 2).

Our smFISH was done in L1s after 5 hours of feeding (but before PGC divisions commenced), so likely post-ZGA. The smFISH analysis of germline genes corroborated RNA-seq and showed that 2 genes that were well expressed in wt PGCs were similarly well expressed in *mes-4* PGCs (see Figure 3 and Figure 3—figure supplement 1). The suggestion to perform time-course smFISH (in late embryos, 30’ fed L1s, and EGCs) would be incredibly labor-intensive, would only sample 1 gene at a time, and is beyond the scope of this study. Furthermore, it is known that some germline genes turn on in PGCs during embryogenesis (e.g. Kawasaki et al., 1998; Subramaniam et al., 1999; Spencer et al., 2011), so the timing of germline gene turn-on spans from embryogenesis to ZGA in PGCs. Nevertheless, we no longer conclude from our transcript profiling analysis that *mes-4* PGCs and EGCs turn on most germline genes normally.

b) The striking high X-linked gene expression in early L1 PGCs and the later observed downregulation of germline genes and upregulation of soma genes in L2 EGCs suggests the possibility that these two regulatory processes may be independent and occur at different times. That is, the upregulation in X-linked gene expression in PGCs might have occurred in embryos, before the major turn on of gene expression in fed L1s, and subsequently there may be a defect in the normal turn on of germline gene expression and repression of somatic gene expression in larvae. The authors should conduct FISH experiments to determine the timing of the different defects.

The up-regulation of X genes in PGCs and later down-regulation of some germline genes and up-regulation of some soma genes in EGCs may be independent or dependent. That would be hard to nail down, even with a handful of smFISH analyses to gauge timing. We note that *mes-4* EGCs up-regulate hundreds more X genes than *mes-4* PGCs, suggesting that the timing of X mis-expression is not restricted to embryogenesis (see Figure 2).

c) The germline and soma gene sets used in this paper are small in number and limit the strength of the conclusions and ability to conclude whether most germline genes turn on normally in mes-4 mutants. The rationale for only assessing a minority of germline expressed genes and for limiting assays to germline genes with germline-specific (n=168) or germline-enriched (n=2178) expression was not clear. To better assess whether PGCs and EGCs have normal turn on of most germline genes and normal repression of somatic genes, the authors should analyse more comprehensive gene lists. To address whether germline genes are turned on normally or not, the authors should assess most germline-expressed genes, including ubiquitously expressed genes. Germline expressed genes in L2 EGCs can be obtained from Cao et al. 2017, from adult germlines in Serizay et al., 2021 and other sources. Genes expressed only in soma can also be found in these and other publications. It should also be tested computationally whether the potential sparseness of the profiling data has limited the ability to observe downregulation of individual germline genes to those with relatively high expression in wild-type. To gain power, the authors could bin germline genes of different expression levels and ask whether expression of the group is lower in mutants.

Yes, see our note in our response to Essential Revisions from the editor. For both PGCs and EGCs, we binned genes by their expression level in wild-type germ cells into deciles and compared gene expression levels between *mes-4* mutant and wild-type germ cells in each decile (see Figure 2—figure supplement 2). Indeed, downregulation was most pronounced in the top decides. This suggests that we lacked sufficient statistical power to call some genes that are low-expressed and have small fold changes down-regulated in *mes-4* PGCs or EGCs. We added a paragraph to the Discussion that speaks to our limited statistical power in identifying down-regulated genes and our application of smFISH as an independent test of differential expression for a few genes. For all genes tested, our smFISH findings corroborated our transcript profiling findings (see Figure 3—figure supplement 1).

To determine whether MES-4 promotes the expression of germline genes, we tested whether the numbers of germline-specific genes and germline-enriched genes in our set of DOWN genes in *mes-4* are higher than expected by chance*.* We chose those sets because they define genes whose products are likely more important to the germline than the soma. We added a more inclusive germline gene set to our analysis: “MES-4-bound genes”, which defines 4132 genes that are bound by MES-4 in embryos, most of which are germline-expressed (Rechtsteiner et al., 2010). Our findings for this gene set are similar to the findings for the germline-specific and germline-enriched sets (see Figures 1 and 2). We do not think it is useful to analyze the enrichment of germline-expressed genes (including housekeeping genes) in our set of DOWN genes in *mes-4*, because every DOWN gene qualifies as a germline-expressed gene (i.e. they are expressed in wild-type PGCs and/or EGCs). To test whether a somatic program is mis-expressed in *mes-4*, we aimed to analyze genes that are expressed in somatic cells and lowly or not expressed in germ cells. We evaluated whether genes in our soma-specific set and in 2 more inclusive “soma” gene sets (Gaydos et al., 2012 and Lee et al., 2017) are not expressed in germ cells (see Author response image 1). We found that few genes in our set are expressed above background levels in PGCs or EGCs. However, many genes in the 2 more inclusive soma sets have high expression levels in our pure samples of dissected PGCs and EGCs. Therefore, while our soma-specific set has fewer genes than the other 2 sets, it more accurately defines soma-specific genes, and we prefer a more accurate than inclusive set for our analysis.

**Author response image 1. sa2fig1:** Evaluation of the specificity of our soma-specific gene set and 2 other “soma” gene sets. (A,B,C,) MA plots showing log2(average expression) versus log2(fold change) of transcript abundance for 20,258 protein-coding genes (circles) between mes-4 and wt PGCs. Numbers of biological replicates (sets of 2 cells): 19 wt PGCs and 11 mes-4 PGCs. Genes that exceed one or both plot scales (triangles) are set at the maximum value of the scale. Genes belonging to a “soma” gene set are colored: (A) 861 soma-specific genes that were defined in this study, (B) 2658 soma genes that were defined in Lee et al. (2017), and (C) 4259 soma genes that were defined in Gaydos et al. (2012). Differentially expressed genes in mes-4 vs wt PGCs were identified using Wald tests in DESeq2 and by setting a q-value < 0.05 significance threshold. Numbers of all mis-regulated genes (black) and numbers of those in the soma gene sets (colored) are indicated in the corners; top is upregulated (UP) and bottom is downregulated. (D) Table showing, for each soma gene set, the number of genes in the set, the number of genes that are expressed in wt PGCs, and the percentage of genes that are expressed in wt PGCs. We categorized 5,858 protein-coding genes that have a minimum average normalized read count of 1 in wt PGCs as “expressed in wt PGCs”.

Gaydos LJ, Rechtsteiner A, Egelhofer TA, Carroll CR, Strome S (2012) Antagonism between MES-4 and Polycomb repressive complex 2 promotes appropriate gene expression in *C. elegans* germ cells *Cell reports* 2:1169–1177. https://doi.org/10.1016/j.celrep.2012.09.019

Lee CS, Lu T, Seydoux G (2017) Nanos promotes epigenetic reprograming of the germline by down-regulation of the THAP transcription factor LIN-15B *eLife* 6:e30201. https://doi.org/10.7554/*eLife*.30201

3. The authors should include tables of all of the gene sets they identified to be up/down in PGCs and EGCs in different mutant backgrounds, i.e., up in mes-4 L1 PGCs, down in mes-4 L1 PGCs,.…

Done. See Supplementary File 1.

4. The GEO submission indicates the availability of a file of normalized counts that I couldn't find: "A matrix of normalized counts for samples of PGCs and EGCs is included in this submission: pgc_egcs_countsNormalized.txt pgc_egcs_countsNormalized.txt"

We fixed that.